# Simultaneous bicolor interrogation in thulium optical clock providing very low systematic frequency shifts

Artem A. Golovizin [1✉], Dmitry O. Tregubov [1], Elena S. Fedorova[1], Denis A. Mishin[1], Daniil I. Provorchenko[1], Ksenia Yu. Khabarova[1,2], Vadim N. Sorokin[1] & Nikolai N. Kolachevsky[1,2]

Optical atomic clocks have already overcome the eighteenth decimal digit of instability and uncertainty, demonstrating incredible control over external perturbations of the clock transition frequency. At the same time, there is an increasing demand for atomic (ionic) transitions and new interrogation and readout protocols providing minimal sensitivity to external fields and possessing practical operational wavelengths. One of the goals is to simplify the clock operation while maintaining the relative uncertainty at a low $10^{-18}$ level achieved at the shortest averaging time. This is especially important for transportable and envisioned space-based optical clocks. Here, we demonstrate implementation of a synthetic frequency approach for a thulium optical clock with simultaneous optical interrogation of two clock transitions. Our experiment shows suppression of the quadratic Zeeman shift by at least three orders of magnitude. The effect of the tensor lattice Stark shift in thulium can also be reduced to below $10^{-18}$ in fractional frequency units. This makes the thulium optical clock almost free from hard-to-control systematic shifts. The "simultaneous" protocol demonstrates very low sensitivity to the cross-talks between individual clock transitions during interrogation and readout.

[1] P.N. Lebedev Physical Institute, Moscow, Russia. [2] Russian Quantum Center, Moscow, Russia. ✉email: artem.golovizin@gmail.com

The excellent applicability of atoms as frequency references is due to the high-quality factor of certain atomic transitions and the possibility of isolating atoms from the environment. The first atomic microwave frequency reference[1] was demonstrated in 1955, and in 1967, the SI second was redefined[2] (9.2 GHz hyperfine transition in Cs). Later, advances in laser technology, manipulation of atoms and ions, and optical frequency measurements paved the way for high-Q optical frequency references[3–5]. Modern optical clocks operating at the $10^{-18}$ level of relative instability and uncertainty[6–14] stimulate discussion about inevitable redefinition of the SI second[15,16], which is also motivated by comparisons of different types of optical clocks at the $10^{-17}$ level[6,7,17,18]. Such performance allows one to accurately test some fundamental physical theories: general relativity[19,20], Lorentz invariance[21,22], drifts of fundamental constants[23,24], and the search for dark matter particles[25] and dark matter clusters[26].

High-performance transportable optical clocks are considered an essential part of a worldwide optical clock network[20,27–29]. Together with stabilized optical fiber links[7,30], transportable systems are requested for long-distance time and frequency comparisons and chronometric leveling[19,20]. The most advanced systems are based on single ions in Paul traps and ensembles of neutral atoms in optical lattices. Single-ion systems are typically less sensitive to the environment and allow simpler setups, while clocks based on ensembles of neutral atoms (typically up to $10^5$) demonstrate better stability. The largest systematic shifts in lattice clocks come from the lattice and interrogation light fields, blackbody radiation (BBR), and external magnetic fields (the Zeeman shift)[19,20,31]. To overcome these difficulties, novel clock transitions are explored[32–35], and operation protocols allowing to significantly reduce the systematic shifts are developed[36–40]. As an example, construction of a "virtual" synthetic frequency was proposed to suppress the BBR shift[36,37] or the quadratic Zeeman shift[38,39].

In this work, we demonstrate advantages of a synthetic frequency approach in thulium optical clock, particularly for canceling out the quadratic Zeeman shift. We implement simultaneous optical interrogation of two clock transitions such that the synthetic frequency is constructed from a single measurement and is highly insensitive to any fluctuations of the magnetic field. Together with other favorable properties of the 1.14 μm inner-shell transition in $^{169}$Tm, this approach allows us to cancel out all main systematic frequency shifts to low $10^{-18}$ level.

## Results

### Synthetic frequency for the 1.14 μm transition in thulium.

The advantageous metrological properties of the 1.14 μm clock transition in $^{169}$Tm are associated with the similarity of the clock levels and their wave functions. The clock levels are the fine-structure components of the same ground-state electronic level separated by an optical interval of 1.14 μm. The transition is associated with a spin-flip in the 4f-electron shell, which is strongly shielded by the outer closed $5s^2$ and $6s^2$ shells (Fig. 1a). Both factors make the transition frequency highly insensitive to a dc external electric field and collisions, which was first experimentally observed in 1983[41]. We previously showed[42,43] that the 1.14 μm ($\nu_0 = 2.6 \times 10^{14}$ Hz) transition possesses very small susceptibility to a dc external electric field, which provides very low sensitivity to blackbody radiation (BBR). At room temperature, the BBR frequency shift corresponds to only $2.3(1.1) \times 10^{-18}$ in fractional units, which is thousands of times smaller than, e.g., that in the strontium lattice clock.

However, the advantages of the 4f-shell clock transition come with a price: thulium atom has a relatively large magnetic dipole moment of $4\mu_B$ in the ground state, where $\mu_B$ is the Bohr

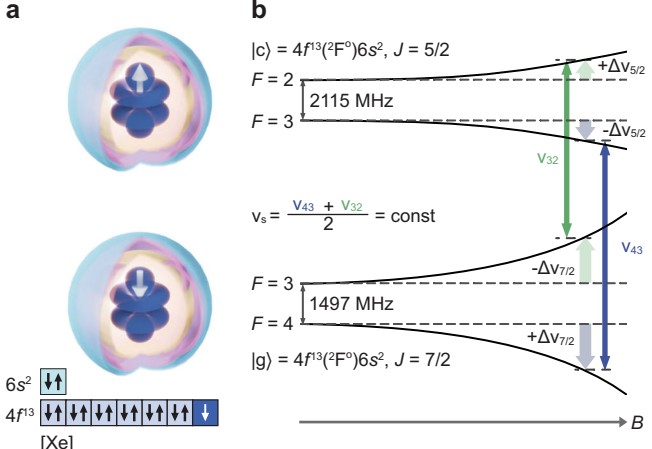

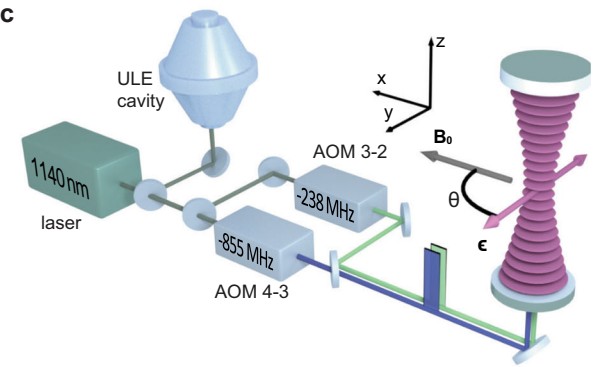

**Fig. 1 Bicolor interrogation scheme in the Tm clock using the hyperfine components of the clock levels. a** Visual representation of the electronic structure of the ground and clock levels. An unpaired 4f-electron is located inside the closed 6s shell. The white arrow represents the spin orientation of the unpaired electron. [Xe] refers to Xe electronic structure. **b** Magnetic doublets of the ground- and excited-state clock levels. Two simultaneously interrogated transition frequencies are denoted as $\nu_{43}$ (blue) and $\nu_{32}$ (green), $\Delta\nu$ is the quadratic Zeeman frequency shift of the corresponding level. $|g\rangle$ and $|c\rangle$ are notations of the ground and clock levels, correspondingly. **c** Sketch of the experimental setup. The 1.14 μm laser is stabilized to the ULE cavity. Two acousto-optical modulators (AOMs) form pulse sequences for simultaneous interrogation of the "4-3" (blue) and "3-2" (green) clock transitions of atoms trapped in the vertically oriented optical lattice at 1064 nm. $\theta$ is the angle between the bias magnetic field $\mathbf{B}_0$ and lattice polarization $\epsilon$.

magneton. The contribution from the dipole–dipole interaction can be readily canceled out by the use of the transition between zero projections of the total atomic momentum $m_F = 0 \rightarrow m_F = 0$. Large orbital momentum results in a nonzero differential tensor polarizability, which is numerically small (0.2 a.u. at 1064 nm) but significant at the desired $10^{-18}$ level of uncertainty. Nevertheless, the key systematic contribution to the frequency shift of the 4f-shell clock transition is the second-order Zeeman shift of 257 Hz G$^{-2}$, which is large compared to other atomic species used in optical clocks. These two effects—the Zeeman effect and the tensor polarizability—in our case account for the main undesired systematic contributions. Addressing these two shifts simultaneously requires an unconventional approach because of their opposite behavior with respect to a bias magnetic field $B_0$. Decreasing the bias magnetic field reduces the quadratic Zeeman shift and corresponding uncertainty, while the uncertainty of the tensor Stark shift generally increases.

To resolve this, we use a simultaneous bicolor interrogation scheme of the two ground-state hyperfine sublevels $F = 4$ and $F = 3$ with the instantaneous synthetic frequency

$$\nu_s = \frac{\nu_{43} + \nu_{32}}{2} \qquad (1)$$

as depicted in Fig. 1b. Here, $\nu_{43}$ and $\nu_{32}$ are the frequencies of the $\left|g, F = 4, m_F = 0\right\rangle \rightarrow \left|c, F = 3, m_F = 0\right\rangle$ (short notation "4-3") and $\left|g, F = 3, m_F = 0\right\rangle \rightarrow \left|c, F = 2, m_F = 0\right\rangle$ ("3-2") transitions, respectively. $\left|g\right\rangle$ stands for the $|4f^{13}(^2F^o)6s^2, J = 7/2\rangle$ ground level, and $\left|c\right\rangle$ stands for the $|4f^{13}(^2F^o)6s^2, J = 5/2\rangle$ clock level. The synthetic frequency $\nu_s$ is found to be insensitive to a magnetic field because of the equal but opposite Zeeman shifts of the "4-3" and "3-2" clock transitions (see the "Methods" section for details). We note here, that recently implemented dynamic decoupling scheme[40,44] allows one to average out the second-order Zeeman shift and quadrupole electric shift.

A synthetic frequency is regularly used in various optical clocks when different magnetic sublevels are interrogated successively[45]. However, the proposed approach, as well as the dynamic decoupling[40] and some other combined and synthetic frequency schemes[38], possess an important advantage. In our configuration, we simultaneously probe two hyperfine clock transitions such that the impact of a fluctuating external magnetic field on the two frequencies becomes completely identical. This fact results in full cancellation of the Zeeman shift without any assumptions about the magnetic field behavior between consecutive measurements. To successfully implement this approach, one should (i) simultaneously prepare an atomic ensemble in two initial states and (ii) independently and simultaneously interrogate and readout two hyperfine clock transitions every measurement cycle.

Simultaneous single-frequency optical pumping to $\left|F = 3, 4, m_F = 0\right\rangle$ central magnetic sublevels was demonstrated in our previous work[46]. In turn, the relatively small separation of the clock transition frequencies of $\Delta\nu = \nu_{32} - \nu_{43} \approx 617$ MHz simplifies simultaneous interrogation and readout using acousto-optical modulators (AOMs), as it is shown in Fig. 1c. At the same time, the ac Stark shift induced by the bicolor probe field is only $0.06\,\mu$Hz under our experimental conditions because of the low probe field intensity. The mutual influence of interrogation and readout fields in the bicolor scheme is discussed in details in the "Methods" section. It is also worth mentioning that the polarizabilities of the $m_F = 0$ sublevels in the hyperfine doublets are identically equal such that the condition for the magic wavelength is fulfilled simultaneously for both clock transitions.

**Zeroing the Zeeman shift in the bicolor scheme**. We prepare $\sim 10^5$ atoms in the $\left|g, F = 4, m_F = 0\right\rangle$ state and $10^4$ atoms in the $\left|g, F = 3, m_F = 0\right\rangle$ state using simultaneous optical pumping[46] with a subsequent lattice depth ramp to sift out hot atoms from the trap. Two clock transitions are simultaneously excited by 80 ms-long clock laser $\pi$-pulses with frequencies $\nu_{43}$ and $\nu_{32}$ separated by $\sim 617$ MHz. Using the readout procedure described in "Methods" section, we simultaneously measure the excitation probability for each of the transitions for certain probe field detunings. We prove that our experimental procedure allows one to independently deduce two excitation efficiencies. The system operates in the classical optical clock regime when the laser frequency is alternately switched between the left and right slopes of the clock transition. In our case, the spectral linewidth of both clock transitions is $\delta\nu = 10$ Hz. Using this method, we deduce two independent error signals and two clock transition frequencies $\nu_{43}$ and $\nu_{32}$. As a stable frequency reference, we use a $1.14\,\mu$m clock cavity with linear drift correction[47]. The impact of an external parameter (e.g., a bias magnetic field $B_0$) on the transition

frequency can be studied by differential measurement when the parameter is alternately changed between two values for odd and even measurement cycles. The multichannel digital locks track the corresponding transition frequencies. Calibration measurements are periodically performed to monitor and tune auxiliary parameters (see "Methods" section for details).

To experimentally verify cancellation of the Zeeman shift for the synthetic frequency $\nu_s$, we change the bias magnetic field between two largely different values: the "reference" field $B_0^r = 218$ mG and the "main" field $B_0^m = 132$ mG. The digital locks track the corresponding frequency shifts with the help of the AOMs (see Fig. 1c). We deduce differential frequency shifts of the two clock transitions of

$$\begin{aligned} \Delta\nu_{43} &= \nu_{43}(B^m) - \nu_{43}(B^r), \\ \Delta\nu_{32} &= \nu_{32}(B^m) - \nu_{32}(B^r) \end{aligned} \qquad (2)$$

and the corresponding synthetic frequency shift

$$\Delta\nu_s = (\Delta\nu_{43} + \Delta\nu_{32})/2. \qquad (3)$$

The frequency differences $\Delta\nu_{43}$, $\Delta\nu_{32}$, and $\Delta\nu_s$ measured for 2300 s are shown in Fig. 2a. The mean value for $\Delta\nu_{43}$ equals $-7.88(9)$ Hz, while for $\Delta\nu_{32}$, it equals $+7.72(7)$ Hz, where the number in parentheses indicates one standard deviation. For the synthetic frequency, the effect averages out to $-0.08(6)$ Hz. Figure 2b shows the Allan deviation for the corresponding data sets. For longer averaging times, we see the $3.5 \times \tau^{-1/2}$ Hz behavior for $\Delta\nu_s$, which corresponds to $1.4 \times 10^{-14}\tau^{-1/2}$ (here, $\tau$ is taken in seconds) in relative units. For short averaging times of 1 s, the Allan deviation approaches the instability of the clock laser of $2 \times 10^{-15}$, which was previously characterized[48]. Fluctuations of the interrogation laser and the performance of the digital locks are discussed in "Methods" section.

Similar measurements were performed for different values of the "main" bias magnetic field $B_0^m$ in the range of $130-520$ mG at fixed $B_0^r = 218$ mG. Corresponding data for $\Delta\nu_{43}$, $\Delta\nu_{23}$, and the synthetic frequency shift $\Delta\nu_s$ are shown in Fig. 3a. The frequency dependencies $\Delta\nu_{43}(B_0^m)$ and $\Delta\nu_{32}(B_0^m)$ are fitted by a parabolic

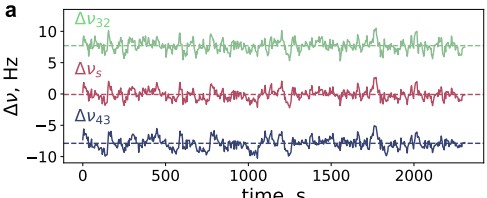

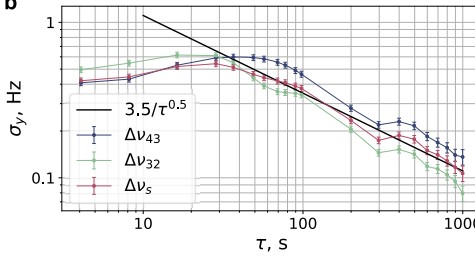

**Fig. 2 Zeeman frequency shift. a** Measurement of clock transition frequency shifts $\Delta\nu_{43} = \nu_{43}(B^m) - \nu_{43}(B^r)$ (blue) and $\Delta\nu_{32} = \nu_{32}(B^m) - \nu_{32}(B^r)$ (green) for two alternately changing values of the bias magnetic field $B_0^r = 218$ mG and $B_0^m = 132$ mG. Common features in the $\Delta\nu_{43}$ and $\Delta\nu_{32}$ data sets come from clock laser frequency fluctuations. The synthetic frequency shift $\Delta\nu_s = (\Delta\nu_{43} + \Delta\nu_{32})/2$ is plotted in red with the mean value of $-0.08(6)$ Hz (dashed line). **b** Allan deviation plotted for each of the data sets. The solid black line indicates the $\sim 1/\sqrt{\tau}$ dependency. Error bars correspond to 1 s.d. statistical uncertainty.

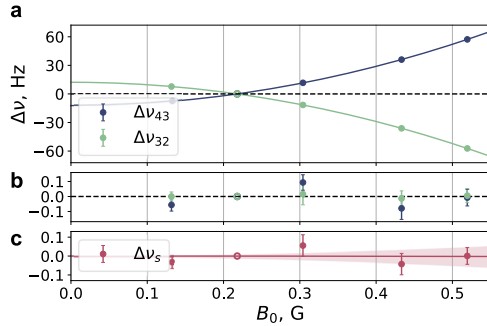

**Fig. 3 Sensitivity of the clock "43" and "32" transitions and synthetic frequency to the bias magnetic field value. a** Frequency shifts $\Delta\nu_{43}$ and $\Delta\nu_{32}$ as a function of the "main" bias magnetic field $B_0^m$ at a fixed value of the "reference" field of $B_0^r = 218$ mG. Solid lines are parabolic fits to the data. **b** Fit residuals. The synthetic frequency shift $\Delta\nu_s$ is shown in (**c**). The shaded area indicates the $1\sigma = 0.175$ Hz G$^{-2}$ uncertainty of the quadratic Zeeman coefficient. Error bars correspond to 1 s.d. statistical uncertainty.

function of the form $\beta(B_0^m - B_0^r)$, where $\beta$ is the quadratic Zeeman coefficient. We obtain $\beta_{43} = +257.47\,(22)(75)$ Hz G$^{-2}$ and $\beta_{32} = -257.42\,(18)(75)$ Hz G$^{-2}$. The value in the first parentheses represents the $1\sigma$ statistical error of the fit. The value in the second parentheses corresponds to the $1\sigma$ uncertainty due to the measurement error of the bias magnetic field (see "Methods" section for details). The absolute values of $\beta_{43}$ and $\beta_{32}$ are equal within the combined uncertainty. The averaged value $\beta = 257.44\,(14)(75)$ Hz G$^{-2}$ agrees with the calculated value[43] of $\beta^{\text{th}} = 257.2$ Hz G$^{-2}$.

The synthetic frequency shift depicted in Fig. 3c is also fitted by a parabolic function, which gives $\beta_s = -0.008(175)$ Hz G$^{-2}$. It is compatible with zero within the measurement uncertainty and corresponds to at least a 1000-fold reduction in sensitivity to the Zeeman shift compared to an individual clock transition.

**Tensor Stark shift from the optical lattice field.** Zeeman shift cancellation opens a way to improve control of the tensor Stark shift. The clock transition frequency shift from the optical lattice is given by

$$\Delta\nu = \Delta\nu^s + \Delta\nu^t = (\tilde{\alpha}^s + \tilde{\alpha}^t \times (3\cos^2\theta - 1)) \times (U/E_r). \quad (4)$$

where $\theta$ is the angle between the bias magnetic field $\mathbf{B}_0$ and the lattice polarization vector $\epsilon$, as shown in Fig. 1c. Here, $U$ is the lattice depth, and $E_r = \hbar^2 k^2/(2m) = h \times 1043$ Hz is the lattice photon recoil energy. The coefficients $\tilde{\alpha}^s$ and $\tilde{\alpha}^t$ are associated with the differential scalar and tensor polarizabilities of the 1.14 $\mu$m transition in Tm at 1064 nm[43]. In our experiment, a 1D optical lattice at 1064 nm is aligned vertically along the $z$ axis, its polarization $\epsilon$ is parallel to the $y$ axis, and $\mathbf{B}_0$ is parallel to the $x$ axis such that the angle $\theta = \pi/2 + \delta\theta$, where $|\delta\theta| \ll 1$ (see Fig. 1c). In this case, Eq. (4) can be rewritten as

$$\Delta\nu = \Delta\nu_{\pi/2} + \delta^t.$$
$$\Delta\nu_{\pi/2} = (\tilde{\alpha}^s - \tilde{\alpha}^t) \times (U/E_r) \quad (5)$$
$$\delta^t = 3\tilde{\alpha}^t(U/E_r) \times \delta\theta^2$$

Here, $\Delta\nu_{\pi/2}$ is the lattice ac Stark shift for $\theta = \pi/2$. At the magic wavelength $\Delta\nu_{\pi/2} = 0$. $\delta^t$ is a $\theta$-dependent part of the polarizability. For the typical trap depth of $U = 100\,E_r$ and $\tilde{\alpha}^t = 0.7$ Hz (see "Methods" section), accurate control of $\delta^t$ below 1 mHz requires $|\delta\theta| \lesssim 10^{-3}$. In our configuration, $\delta\theta = B_y/B_0$ assuming that the direction of the lattice polarization is much more stable than the direction of the external magnetic field. To reduce $\delta^t$, it is necessary to increase the bias magnetic field $B_0$.

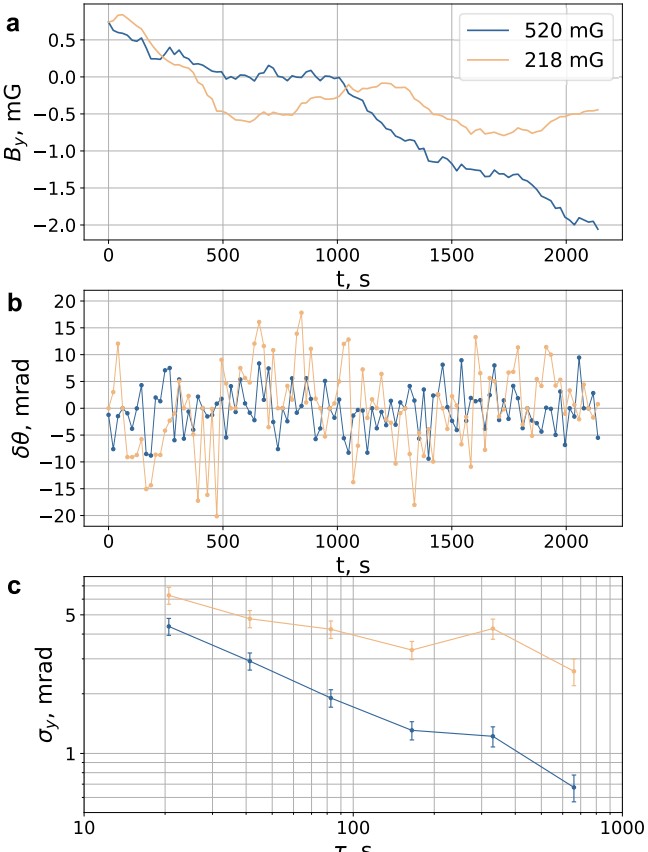

**Fig. 4 Stabilization of the bias magnetic field direction. a** Compensation field $B_y$ as a function of time for one measurement run. The orange line corresponds to the data points with the "reference" field $B_0^r = 218$ mG and the blue line to the points with the "main" field $B_0^m = 520$ mG. **b** Evaluated $\delta\theta$ over the measurement run. **c** Allan deviation for $\delta\theta$ data sets from (**b**). Error bars correspond to 1 s.d. statistical uncertainty.

Using the bicolor scheme, we actively stabilize $\theta$ to $\pi/2$ using the following method. During a typical measurement run, we perform perpendicular magnetic field calibration every 5th cycle. Clock transition frequency shifts $\delta_\pm^t$ are measured for two opposite signs of the deliberately added/subtracted increment $\Delta\theta$, which gives $\theta = \pi/2 + \delta\theta \pm \Delta\theta$. The increment $\Delta\theta \approx 0.1$ is added by changing the perpendicular magnetic field $B_y$. This method significantly increases the sensitivity of the frequency shift from the $\sim\delta\theta^2$ dependency to $\sim\delta\theta\Delta\theta$. The misalignment $\delta\theta$ is deduced from the frequency mismatch $\delta_+^t - \delta_-^t$ (see "Methods" section for details). Note that fluctuations of the magnetic field along the $z$ axis do not influence the angle $\theta$.

Figure 4a shows traces of the perpendicular magnetic field $B_y$ produced by the compensation coils. For the given run, we apply $B_0^r = 218$ mG and $B_0^m = 520$ mG. Depending on the value of the bias field ("reference" or "main"), different values of the compensation field are applied, which are indicated as orange and blue lines, respectively. The plots in Fig. 4b show the misalignment angle $\delta\theta$, which is deduced from corresponding magnetic field measurements. The corresponding Allan deviation for $\delta\theta$ data sets is shown in Fig. 4c. For the larger bias magnetic field $B_0^m = 520$ mG, the fluctuations of $\delta\theta$ are smaller. In this case, an uncertainty of $\delta\theta = 1$ mrad is achieved after 500 s of integration. For $B_0^r = 218$ mG, the averaging proceeds slower due to the larger shot-to-shot fluctuations. Using these results, one can evaluate the contribution of the tensor Stark shift from the optical lattice (the lattice depth $U = 300\,E_r$) using the Eq. (5). We obtain

corresponding contributions to the fractional frequency uncertainty of $\delta^{t}/\nu_0 < 10^{-17}$ for $B_0^{r} = 218\,\text{mG}$ and $\delta^{t}/\nu_0 < 10^{-18}$ for $B_0^{m} = 520\,\text{mG}$ after 500 s of integration. Note that an increase in the bias magnetic field does not lead to additional Zeeman shift-associated uncertainty because of the bicolor operation scheme.

## Discussion

In this work, we showed that simultaneous bicolor optical interrogation of two clock transitions and construction of the synthetic frequency leads to at least 1000-fold suppression of the quadratic Zeeman shift in thulium optical lattice clock. Our analysis shows the following important aspects of this scheme. First, the synthetic frequency is completely insensitive to any fluctuations of the magnetic field during simultaneous interrogation. This significantly softens requirements on the magnetic shielding providing substantial advantages for the transportable setups. Second, simultaneous interrogation does not lead to any additional systematic frequency shift of either individual or synthetic clock frequencies. Moreover, acquiring information about two clock transitions simultaneously in the single cycle reduces the instability of synthetic frequency by 1.4 times compared to the regular sequential scheme.

Considering the thulium optical clock, cancellation of the Zeeman frequency shift allowed us to implement control over the tensor Stark shift induced by the optical lattice at the $10^{-18}$ level by accurate online tuning of the magnetic field direction. Together with results from ref. [43] demonstrating the very low sensitivity of the Tm clock transition to the BBR shift ($2.3 \times 10^{-18}$ at room temperature), the suggested bicolor Tm optical clock becomes almost free from systematic frequency shifts at the $10^{-18}$ level. The operational magic wavelength of ~1064 nm promises further advantages compared to many other optical lattice clocks. Besides availability of powerful and low-noise fiber lasers, it provides small sensitivity of the clock transition frequency to the lattice wavelength (1000 times smaller than that of the Sr optical clock), while the influence of the high-order lattice shifts are similar. Thus, a bicolor Tm optical clock operating at the 1.14 μm inner-shell magnetic dipole transition demonstrates functional capacity for the next generation of transportable systems with very moderate requirements for environmental conditions, temperature, and magnetic field fluctuations.

## Methods

**Zeeman shift and synthetic frequency.** The energy of an atomic level with electronic momentum $J$ and nuclear spin $I = 1/2$ in an external magnetic field $B_0$ is given by[42,49]

$$E_{J,F=J\pm1/2,m_F} = -\frac{1}{4}hA_J + g_J\mu_B B_0 m_F$$
$$\pm \frac{hA_J(2J+1)}{4}\sqrt{1 - \frac{4m_F}{2J+1}x + x^2},$$
(6)

where $A_J$ is the hyperfine splitting constant, $x = \frac{2(g_J\mu_B - g_I\mu_N)B_0}{hA_J(2J+1)}$, $g_J$ and $g_I$ are the electronic and nuclear Landé g-factors, and $\mu_B$ and $\mu_N$ are the Bohr and nuclear magnetons, respectively.

For the magnetic sublevel with zero projection $m_F = 0$ and $x \ll 1$, the level energy shift can be expressed as

$$\Delta E_{J,F=J\pm1/2} = \pm\frac{(g_J\mu_B - g_I\mu_N)^2}{2hA_J(2J+1)}B_0^2 = \mp\beta_J B_0^2,$$
(7)

where $\beta_J$ is the quadratic Zeeman shift coefficient for a given electronic level. The negative sign of $A_J$ for the ground and clock levels in thulium is taken into account. The coefficients $\beta_{7/2} = 426\,\text{Hz}\,\text{G}^{-2}$ and $\beta_{5/2} = 169\,\text{Hz}\,\text{G}^{-2}$ are relatively large due to the small (1.5 and 2.1 GHz, respectively) hyperfine splitting of the ground and clock levels. The frequencies of the $|g, F=4, m_F=0\rangle \rightarrow |c, F=3, m_F=0\rangle$ and $|g, F=3, m_F=0\rangle \rightarrow |c, F=2, m_F=0\rangle$ clock transitions as a function of $B_0$ can be

found as follows:

$$h\Delta\nu_{43} = \Delta E_{5/2,3} - \Delta E_{7/2,4} = -(\beta_{5/2} - \beta_{7/2})B_0^2,$$
$$h\Delta\nu_{32} = \Delta E_{5/2,2} - \Delta E_{7/2,3} = (\beta_{5/2} - \beta_{7/2})B_0^2.$$
(8)

Thus, for the synthetic frequency

$$\Delta\nu_s = \frac{\Delta\nu_{43} + \Delta\nu_{32}}{2},$$
(9)

the quadratic Zeeman shift from the hyperfine interaction completely cancels out. This result remains valid even without the Taylor expansion of Eq. (6). A small nonzero quadratic coefficient of the synthetic frequency of the clock transition may appear from coupling of the clock levels to other atomic levels. The major contribution to this coefficient is due to coupling of the clock levels themselves, which are the fine structure levels of the ground electronic state. Assuming the LS coupling and using formula similar to Eq. (6), the quadratic coefficient $\beta_{1.14} = 3.8\,\text{mHz}\,\text{G}^{-2}$. At $B_0 = 0.218\,\text{G}$, the relative frequency shift is estimated as $\Delta\tilde{\nu}_{\beta_{1.14}} = 0.7\times10^{-18}$.

**Tensor polarizability coefficient.** Following ref. [43], the absolute value of the optical lattice trap depth can be expressed as

$$U = \left|-\alpha\frac{\mathcal{E}^2}{4}\right|,$$
(10)

where $\mathcal{E}$ is the amplitude of the electric field and $\alpha = 152$ a.u. (atomic units) is the ground-level polarizability. The clock transition frequency shift from the tensor part of the polarizability is

$$h\Delta\nu^{t} = -\frac{3\cos^2\theta - 1}{2}\Delta\alpha^{t}\frac{\mathcal{E}^2}{4}.$$
(11)

Here $\Delta\alpha^{t} = -0.2$ a.u.[43] for the lattice wavelength of 1064 nm. Inserting Eq. (10) into Eq. (11) and rearranging it, one obtains

$$\Delta\nu^{t} = -(3\cos^2\theta - 1)\times\frac{U}{E_r}\times\frac{E_r}{2\alpha h}\Delta\alpha^{t}.$$
(12)

Comparing this equation to Eq. (4) and using $E_r = h \times 1043\,\text{Hz}$, we find

$$\tilde{\alpha}^{t} = -\frac{E_r}{2h}\frac{\Delta\alpha^{t}}{\alpha} = 0.7\ \text{Hz}.$$
(13)

**Excitation efficiency and readout procedure.** After simultaneous excitation of two clock transitions, we implement a dedicated readout procedure to deduce the excitation efficiency of each of the transitions. Every readout is destructive, i.e., all measured atoms are removed from the trap. Below, we use the notation $|b\rangle$ for the $|4f^{12}(^3H_5)5d_{3/2}6s^2, J = 9/2\rangle$ level, which is used for the first-stage laser cooling. The relevant thulium levels are shown in Fig. 5a. The sequence of readout pulses is shown in Fig. 5b.

The readout procedure aims to measure the populations of the four hyperfine sublevels $|g, F=4\rangle$, $|g, F=3\rangle$, $|c, F=3\rangle$ and $|c, F=2\rangle$ shown in Fig. 5a. The pulse sequence is as follows:

1. With the help of a 0.2 ms resonant probe 4-5 pulse ($|g, F=4\rangle \rightarrow |b, F=5\rangle$), we measure the number of atoms remaining in the $|g, F=4\rangle$ state after excitation by the "4-3" clock pulse, which is denoted as $n_{g,4}$.
2. Through two overlapping 0.7 ms resonant probe 4-5 and 0.5 ms probe 3-4 ($|g, F=3\rangle \rightarrow |b, F=4\rangle$) pulses, we determine the number of atoms $n_{g,3}$ remaining in the $|g, F=3\rangle$ state. The $|g, F=3\rangle \rightarrow |b, F=4\rangle$ pulse repumps atoms from the $|g, F=3\rangle$ to $|g, F=4\rangle$ state.
3. Two consecutive $\pi$-pulses, clock 4-3 (1 ms duration) and clock 3-2 (4 ms duration), return atoms from the $|c, F=3\rangle$ to $|g, F=4\rangle$ states and from the $|c, F=2\rangle$ to $|g, F=3\rangle$ states, respectively.
4. Similar to p.1, we measure $n_{c,3}$, which is proportional to the number of atoms excited by the clock 4-3 pulse.
5. Similar to p.2, we measure $n_{c,2}$, which is proportional to the number of atoms excited by the clock 3-2 pulse.

To accurately determine the excitation efficiencies, we deduce the population of the corresponding clock levels as follows:

$$\begin{aligned}
\tilde{n}_{g,4} &= n_{g,4}, \\
\tilde{n}_{c,3} &= \xi_{c3} \times n_{c,3}, \\
\tilde{n}_{g,3} &= n_{g,3} - \xi_{g3} \times n_{c,3}, \\
\tilde{n}_{c,2} &= \xi_{c2} \times n_{c,2}.
\end{aligned}$$
(14)

Here, the coefficients $\xi_{c3}$ and $\xi_{c2}$ are introduced to account for nonideal $\pi$-pulses and spontaneous decay during the readout time from the clock $|c\rangle$ to ground $|g\rangle$ states. The coefficient $\xi_{g3}$ takes into account the influence of spontaneous decay from $|c, F=3\rangle$ to $|g, F=3\rangle$ and $|g, F=4\rangle$ during the time interval between the first and second readout "CMOS" pulses (see Fig. 5b). Note that the population of the $|c, F=2\rangle$ level does not affect the population of the $|g, F=4\rangle$ level because the

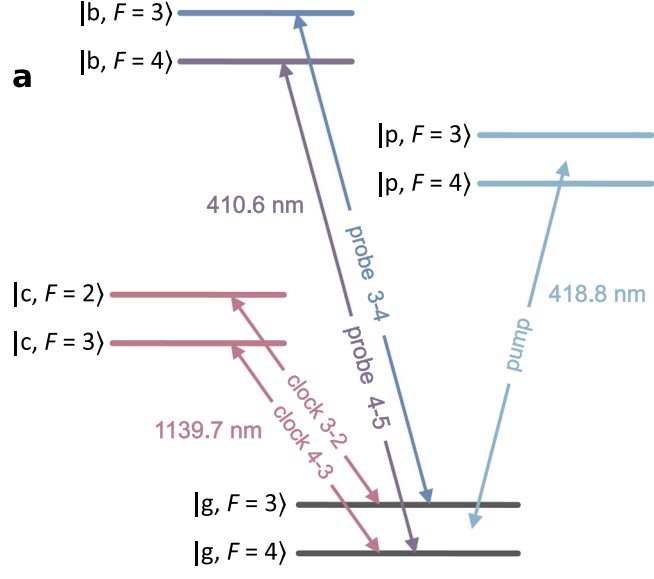

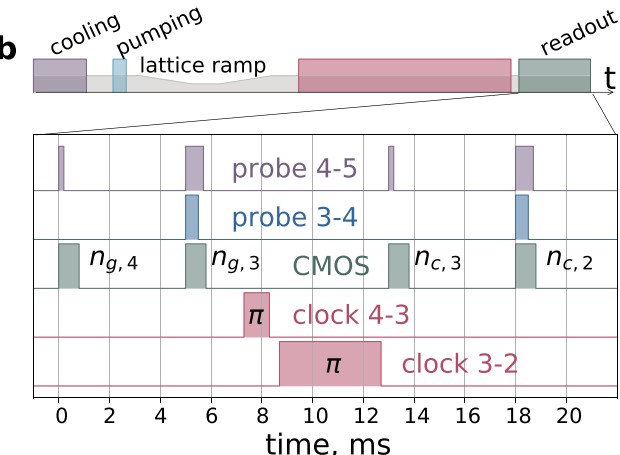

**Fig. 5 Readout protocol. a** Relevant thulium level scheme. The "pump" radiation is detuned by −175 MHz from the $|g, F = 4\rangle \rightarrow |p, F = 4\rangle$ transition and by −90 MHz from the $|g, F = 3\rangle \rightarrow |p, F = 3\rangle$ transition. $|c\rangle$ denotes upper clock level, $|b\rangle$ denotes upper cooling (410 nm transition) level. **b** Measurement cycle (top) and detailed pulse sequence of the readout procedure (bottom). The "CMOS" line shows the exposure periods of the CMOS camera accumulating the fluorescence signal at 410 nm. Labels $n_{\cdot,\cdot}$ indicate the signal recorded in the particular camera image. The clock 4-3 and clock 3-2 pulses are the $\pi$-pulses that transfer the population from the upper clock levels to the ground levels.

transition between them is forbidden. Coefficients $\xi_{g3}$, $\xi_{c3}$, and $\xi_{c2}$ are determined from the condition that the total number of atoms associated with each transition $\tilde{n}_{g,4} + \tilde{n}_{c,3}$ and $\tilde{n}_{g,3} + \tilde{n}_{c,2}$ must not depend on the detuning of any of the clock pulses. Using this condition, the coefficients can be determined from the individual scans of the "4-3" and "3-2" clock transitions shown in Supplementary Fig. 1a, b.

Finally, the excitation probabilities of each of the clock transitions $\eta_{43}$ and $\eta_{32}$ can be calculated as

$$\eta_{43} = \frac{\tilde{n}_{c,3}}{\tilde{n}_{g,4} + \tilde{n}_{c,3}} = \frac{\xi_{c3}\, n_{c,3}}{n_{g,4} + \xi_{c3}\, n_{c,3}} \,, \tag{15}$$

$$\eta_{32} = \frac{\tilde{n}_{c,2}}{\tilde{n}_{g,3} + \tilde{n}_{c,2}} = \frac{\xi_{c2}\, n_{c,2}}{n_{g,3} - \xi_{g3}\, n_{c,3} + \xi_{c2}\, n_{c,2}} \,. \tag{16}$$

**Measurement of the bias magnetic field $B_0$.** The magnitude of the bias magnetic field $B_0$ is deduced from frequency measurements of the two transitions $|g, F = 4, m_F = 0\rangle \rightarrow |c, F = 3, m_F = \pm 1\rangle$ possessing strong first-order Zeeman

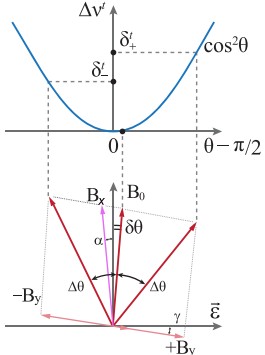

**Fig. 6 Sketch of the bias magnetic field direction stabilization.** Top: clock 4-3 transition frequency shift as a function of the misalignment angle from $\theta_0 = \pi/2$. $\delta_+^t$ ($\delta_-^t$) are the clock transition frequency shifts when additional magnetic field $+B_y$ ($-B_y$) is applied. Bottom: orientation of the bias magnetic field for nominal operation (small deviation $\delta\theta$ from $\theta_0 = \pi/2$) and for calibration measurements $\pm\Delta\theta$. Rotation of $\mathbf{B}_0$ is accomplished by applying additional $\pm B_y$ during the clock interrogation period. The misalignment angle $\delta\theta$ is deduced from the frequency difference $\delta_\pm^t$ according to Eq. (18). Angles $\alpha$ and $\gamma$ depict misalignment of $B_x$ (magnetic field produced by main coils) from $\pi/2$ and $B_y$ from 0, correspondingly.

sensitivity to the magnetic field. We use the formula

$$B_0 = h\frac{\nu^+ - \nu^-}{2g_F\mu_B} \,, \tag{17}$$

where $\nu^+$ and $\nu^-$ are the frequencies of the $\sigma^+$ ($|m_F = 0\rangle \rightarrow |m_F' = 1\rangle$) and $\sigma^-$ ($|m_F = 0\rangle \rightarrow |m_F' = -1\rangle$) transitions, respectively, and $g_F = 0.7121$ is the Landé g-factor of the $|c, F = 3\rangle$ level. In the experiment, we successively record spectra of the $\sigma^+$ and $\sigma^-$ transitions and determine the $\nu^+$ and $\nu^-$ frequencies (with respect to the frequency of the ULE cavity mode) from the approximations of the corresponding spectra with a Gaussian lineshape. The typical frequency shift $\nu^+ - \nu^-$ in our experiments is in the range from 140 to 500 kHz. The observed FWHM of the $\sigma^\pm$ transitions is ~2 kHz, which can be attributed to the fluctuations of the laboratory magnetic field. The typical statistical uncertainty of the line center determination from the fit is 200 Hz. From Eq. (17), we estimate the net uncertainty of the magnetic field determination to be $\sigma_{B_0} = h\frac{\sqrt{2}\delta\nu}{2g_F\mu_B} = 0.14$ mG.

**Stabilization of the magnetic field direction.** To minimize the sensitivity of the clock transition frequencies (both 4-3 and 3-2) to the direction of the bias magnetic field $\mathbf{B}_0$, we chose the angle $\theta = \pi/2$ (see Figs. 1 and 6). According to Eq. (5), the sensitivity to angular deviation $\delta\theta$ at $\theta = \pi/2$ is quadratic. Variations in the laboratory magnetic field can change the angle $\theta$, so one needs to implement active stabilization. We use a pair of coils producing the field component $B_y$, which compensates for the corresponding component of the laboratory field.

To minimize the deviation $\delta\theta$, we apply the following procedure individually for each of the magnetic field values ($B_0^m$ and $B_0^r$). The procedure is repeated after 5 regular measurement cycles. We intentionally introduce an additional angle $\Delta\theta \approx 0.1$ (the absolute value is not important here) between the field direction and the lattice polarization axis ($\boldsymbol{\epsilon} \| \boldsymbol{y}$) by changing $B_y$ at a constant value of $B_x$ (Fig. 6). To change $B_y$, we add/subtract a fixed current increment $\Delta I_y$ to the current flowing through the compensation coils. Two transition frequencies $\nu_+^t$ and $\nu_-^t$ (for each of the fields $B_0^m$ and $B_0^r$) are measured for $\theta = \pi/2 + \delta\theta \pm \Delta\theta$ using a standard procedure of interrogating the clock 4-3 transition on the left and right slopes. The frequency shifts $\delta_\pm^t = \nu_\pm^t - \nu_{43}$ are calculated with respect to the nominal transition frequency $\nu_{43}$ at $\theta = \pi/2 + \delta\theta$ measured in the last measurement cycle. Using Eq. (5), $\delta\theta$ can be found as follows:

$$\delta\theta = \frac{\delta_+^t - \delta_-^t}{12\tilde{\alpha}^t\Delta\theta\,(U/E_r)}. \tag{18}$$

For $\Delta\theta = 0.1$ and $U = 300\,E_r$, the target value of $|\delta\theta| = 10^{-3}$ is reached for the modest uncertainty of $|\delta_+^t - \delta_-^t| = 0.4$ Hz, which, with our current setup, can be achieved in <500 cycles of measurement (see Fig. 4c). The single-measurement value $\delta\theta$ from Eq. (18) is used as an error signal for the proportional-integration digital lock that tunes the offset value of $B_y$ (see Fig. 4a). Thus, the absolute value of $\Delta\theta$, as well as $\alpha^t$ and $U/E_r$, only affects the magnitude of the error signal.

The error in $\theta$ stabilization to $\theta_0 = \pi/2$ may arise from two factors: (i) non-equality of "+" and "−" steps, and (ii) geometrical misalignment of magnetic coils with respect to the lattice polarization $\boldsymbol{\epsilon} \| \boldsymbol{y}$. The former leads to an error $\delta\theta_\pm = \delta B_y/B_0 = \delta B_y/B_y \times B_y/B_0$. In the present setup, we use one channel of the LTC2662 current source (the five-channel, 300 mA Current-Source-Output 12-Bit DACs

from "Analog Devices") to produce $B_y$. Its nonlinearity is <1 mA, hereby for a typical current step size of 100 mA $\delta B_y/B_y \leq 0.01$. Taking into account that $B_y/B_0 \approx 0.1$, we get $\delta\theta_\pm \leq 10^{-3}$, that is within our target uncertainty. In future, we plan to perform "+" and "−" steps by changing polarity of a separate coil, which would allows us to keep the current magnitude constant and use no assumptions about the linearity of the current source.

To evaluate error in $\theta$ stabilization due to the coils misalignment, we consider the following geometry (see Fig. 6): $\mathbf{B}_x$ has an angle of $\pi/2 + \alpha$ to the optical lattice polarization (ideally it is $\pi/2$), and $\mathbf{B}_y$ has an angle of $\gamma$ to the optical lattice polarization (ideally it is 0). Writing down the clock transition frequencies for "+" and "−" $\Delta B_y$ steps (tensor optical lattice shift and second-order Zeeman shift), we find that equilibrium angle $\delta\theta_{eq}$ (i.e., the angle to which the stabilization process converges) equals

$$\delta\theta_{eq} = \gamma\frac{-\eta}{1-\eta}, \quad (19)$$

where $\eta = \frac{\beta B_0^2}{3\tilde{a}^t(U/E_r)}$. In our experiments, $\eta \leq 0.1$ for typical $U = 300\,E_r$ and $B_0 = 0.5$ G (maximum bias magnetic field among the measurements). Maximum misalignment of $\mathbf{B}_y$ and the lattice polarization can be estimated for no more that $\gamma = 0.1$ rad, which can be readily detected by eyes. In this case, we have an error of no more than 0.01 rad, which is worse than we expected. In future experiments, this issue can be addressed from two directions: (i) better alignment of $\mathbf{B}_y$, and (ii) use of not only 4-3 transition for $B_0$ direction calibration, but both 4-3 and 3-2. Indeed, for 3-2 transition the only difference in Eq. (19) is "−" sign of $\beta$, thus

$$\delta\theta_{eq}^{comb} = \frac{\delta\theta_{eq}^{4-3} + \delta\theta_{eq}^{3-2}}{2} = \gamma\frac{-\eta^2}{1-\eta^2}. \quad (20)$$

For the same $\eta = 0.1$ this already gives $\delta\theta_{eq}^{comb} = 0.01 \times \gamma$, that is less than targeted accuracy $\delta\theta = 10^{-3}$. Another possibility is to perform calibration measurements at larger optical trap depth. This would reduce $\eta$ and lead to smaller $\delta\theta_{eq}$. Misalignment of the magnetic field in $z$ direction at the level of 0.1 rad does not effect the alignment procedure described above.

**Magic wavelength determination.** In order to find the magic wavelength one needs to measure frequency shift of the clock transition at different lattice trap depths. As we showed above, we can stabilize magnetic field direction to $\theta = \pi/2$ (angle between the bias magnetic field and lattice polarization direction) with less than 1 mrad error. Since $\theta$ does not depend on the lattice trap depth, the lattice Stark shift would be measured in the same configuration at low and high depths, thus giving the magic wavelength for $\theta = \pi/2$ with error from $\theta$ uncertainty of $<10^{-18}$.

**Nonlinear shifts in the optical lattice.** To estimate ac Stark shift of the clock transition frequency from the optical lattice, one has to consider nonlinear (with intensity I) shifts. They arise from (i) motion of an atom in the lattice potential, particularly along the optical lattice axis, and (ii) from higher-order terms of the polarizability and hyperpolarizability. The straightforward solution to (i) is preparation of atoms in the $n_z = 0$ motional quantum state, which is commonly done in other neutral atoms optical clocks.

Following ref. [50], the lattice Stark shift can be expressed as

$$h\Delta\nu_{LS}(u, \delta_L, n_z) = \left(\frac{\partial\tilde{\alpha}^{E1}}{\partial\nu}\delta_L - \tilde{\alpha}^{qm}\right)\left(n_z + \frac{1}{2}\right)u^{1/2}$$
$$- \left(\frac{\partial\tilde{\alpha}^{E1}}{\partial\nu}\delta_L + \frac{3}{2}\tilde{\beta}\left(n_z^2 + n_z + \frac{1}{2}\right)\right)u \quad (21)$$
$$+ 2\tilde{\beta}\left(n_z + \frac{1}{2}\right)u^{3/2} - \tilde{\beta}u^2.$$

Here $u = U/E_r$ ($U$ is the trap depth), $\delta_L$ is the lattice laser frequency detuning for the E1 magic frequency (wavelength), $\tilde{\alpha}^{E1} = \Delta\alpha^{E1}E_r/\alpha^{E1}$, $\tilde{\alpha}^{qm} = \alpha^{qm}E_r/\alpha^{E1}$ and $\tilde{\beta} = \Delta\beta/(E_r/\alpha^{E1})^2$. The values of $(1/h)\partial\tilde{\alpha}^{E1}/\partial\nu$, $\tilde{\alpha}^{qm}/h$ and $\tilde{\beta}/h$ for 1.14 μm clock transition in Tm and 698 nm transition in Sr (for comparison) are listed in Supplementary Table 1.

For thulium, the polarizability slope was estimated from the measured values at 1064 and 1070 nm. The value of $\alpha^{qm}$ was calculated accounting only for the contribution from M1 transition at 1.14 μm, contributions from other M1 and E2 transitions estimated to be smaller. Thulium hyperpolarizability $\beta$ was calculated as discussed in ref. [42].

**Digital lock performance.** Frequency locking of each of the interrogating light fields to the corresponding clock transition is performed by independent tuning of two AOM frequencies, as shown in Fig. 1c. For every measurement cycle, we simultaneously excite both clock transitions by bicolor radiation with each component alternately detuned by $+\delta\nu/2$ or $-\delta\nu/2$ from the central frequency of the corresponding transition. Here, $\delta\nu$ is the measured transition linewidth, which in our experiments is equal to 10 Hz. It corresponds to the Fourier spectral width of an 80-ms interrogation pulse. To measure the Zeeman shift at two different magnetic fields $B_0^r$ and $B_0^m$, as described in the main text, we change the magnetic

field after probing the excitation efficiencies on the left and right slopes (two consecutive cycles). Parallel digital lock channels are responsible for the two magnetic field values. The integral time constant of the digital proportional-integrating feedback loops equals 100 s.

As a test, we measured the response of the digital locks to an instantaneous change of the bias magnetic field $B_0 = 218$ mG by a small value, as shown in Supplementary Fig. 2. At $t = 190$ s, $B_0$ was increased by $\Delta B_0 = 13$ mG, and at $t = 700$ s, $B_0$ was returned to its original value. The two transition frequencies are shifted by the same amount in the opposite directions, which corresponds to the estimated Zeeman shift. The typical response time of the digital locks is 10 s.

**Mutual influence of the two clock transitions: excitation and readout.** A mutual influence of the clock transitions may occur either (i) from the clock laser fields during the interrogation or (ii) during the readout procedure.

The first effect (i) is associated with the ac Stark shift induced by the other interrogation field and equals $\delta\nu^{AC} \approx \Omega^2/(4\pi^2\Delta\nu_{sep}) = 0.06$ μHz for an 80 ms π-pulse with Rabi frequency $\Omega$ and frequency difference $\Delta\nu_{sep} = 617$ MHz. Other line pulling effects, including quantum interference, are also negligible.

The second effect (ii) results from the population transfer between two pairs of clock levels and is much more pronounced for the 3-2 transition. First, the $|g, F = 3, m_F = 0\rangle$ level is initially 10 times less populated than $|g, F = 4, m_F = 0\rangle$. Second, the upper level $|c, F = 3, m_F = 0\rangle$ of the 4-3 transition decays to $|g, F = 3\rangle$ with a 1/28 probability, while the transition from $|c, F = 2, m_F = 0\rangle$ to $|g, F = 4\rangle$ is forbidden (Fig. 5a). The effect of population transfer is clearly seen in the raw data in Supplementary Fig. 1a–d. However, if the timing of the excitation and readout procedures remains unchanged, then this effect is proportional to the number of atoms excited to the $|c, F = 3, m_F = 0\rangle$ level and can be eliminated using Eq. (14).

To verify this assumption, we recorded frequency scans of two resonances with strong overlap (c,g) and small overlap (d,h), as shown in Supplementary Fig. 1. Generally, the scans of both transitions are independent and are defined by two frequency detunings from the certain 1.14 μm cavity mode. In this experiment, we add a small fixed frequency offset between the two interrogation laser fields, which defines the corresponding scan windows and relative positions of resonances with respect to the net frequency difference of ≈617 MHz. This small offset impacts the relative population of the upper clock levels $|c, F = 2\rangle$ and $|c, F = 3\rangle$ immediately after the interrogation procedure. In both cases, the spectral profiles of the excitation probability are successfully recovered.

The good agreement of the quadratic Zeeman frequency shift measured at different magnetic field values (Fig. 3) also indicates that the readout procedure correctly recovers spectral profiles of the corresponding clock transitions.

To quantitatively evaluate the possible frequency shift from the readout procedure due to the population transfer between the levels, we numerically simulated the digital lock performance. We generated a set of clock laser frequency signals with random frequency walk and white phase noise similar to the noise pattern of our 1.14 μm clock laser. The white phase noise broadens the spectral linewidth to ~5 Hz, and the long-term random frequency walk is usually within 100 Hz during a day. We repeated simulations for different frequency noise levels, including levels considerably higher than those in the experiment. The simulation of the digital lock performance in the presence of laser frequency fluctuations is shown in Supplementary Fig. 3b together with experimental data (Supplementary Fig. 3a). In addition to the intervals $\Delta\nu_{43}$ and $\Delta\nu_{32}$, we show all four transition frequencies at two bias magnetic fields $B_0^r = 218$ mG and $B_0^m = 132$ mG for each of the transitions. The data come from four corresponding digital lock channels. The experimental and simulated traces show very similar behavior. The strong correlation between all four channels indicates that the laser frequency fluctuations are mainly responsible for the short-term instability. The comparison of two Allan deviation plots for the experimental data is shown in Supplementary Fig. 3c also demonstrates a 2.8-times higher instability of the synthetic frequency $\Delta\nu_s = (\Delta\nu_{43} + \Delta\nu_{32})/2$ than of the differential frequency $\Delta\nu_{diff} = (\Delta\nu_{43} - \Delta\nu_{32})/2$. Laser frequency noise should be strongly suppressed in $\Delta\nu_{diff}$, which is proven by the data analysis. In turn, magnetic field fluctuations do not significantly contribute to the instability on the given level.

We introduced the following fluctuations in the model:

1. Quantum projection noise using binomial distribution of the number of atoms (≈$10^4$ initially in $|g, F = 4, m_F = 0\rangle$ and ≈$10^3$ in $|g, F = 3, m_F = 0\rangle$ states) at each step during each readout in the sequence.
2. Clock laser frequency noise: random-walk for frequency and white-noise for phase, similar or larger than experimentally observed.
3. Clock laser power noise: random walk or white phase noise, around 10% of total power, similar or larger than experimentally observed.
4. Fluctuations of the number of atoms in the lattice, dispersion equals 10% of total number.
5. Error in determination of the coefficients $\xi_{c3}$, $\xi_{g3}$, and $\xi_{c2}$ of population transfer, 10% dispersion.
6. Magnetic field fluctuations with different spectra.

Data in Supplementary Fig. 4a, b show histograms of 4-3 and 3-2 clock transitions frequency shifts obtained from 1000 simulation runs. All above-listed fluctuations (except magnetic field noise) were introduced in these simulations. We observe no shift either for 3-2 or 4-3 transitions. The scatter plots

shown in Supplementary Fig. 4c, d indicate that for the majority of the runs (78%, green regions) the inferred frequency shift is smaller than the final Allan deviation.

Regarding the frequency instability, the simultaneous interrogation scheme provides instability of 1.4 times smaller compared to the sequential one, i.e., when in first 2 cycles we interrogate left and right slope of 4-3 transition, and in next 2 cycles we interrogate left and right slope of 3-2 transition. The proportional and integral gains were adjusted for each locking scheme to achieve optimal performance. We observed a factor of 1.4 ($\sqrt{2}$) higher instability for the sequential scheme, which agrees with the one expected from the 2-times decrease of the measurement cycle time. Indeed, for the sequential measurements, we spend only half of the time to probe each of the transitions.

To study the impact of the magnetic field fluctuations we performed simulations for different power spectra of the magnetic field noise. For the brown noise and slow drifts we did not observe any significant difference compared to the other noise sources listed above (no systematic shifts, increase of the Allan deviation). However, if the magnetic field oscillates with the period equals to two full measurement intervals, we observe a systematic frequency shift for the sequential scheme (see Supplementary Fig. 5). This effect readily follows from simple considerations, as in this case both transitions will be shifted in one direction which will not be compensated after building up the synthetic frequency. On the other hand, for the simultaneous method this effect completely cancels out. This is also true for magnetic noise frequencies that are odd-multiple of the full measurement cycle rate. The real spectrum of magnetic frequency noise may and probably does contain these harmonics. The simultaneous interrogation method offers straightforward solution of this envisioned problem.

## Data availability
All data are available within the Article and Supplementary Files, or available from the corresponding author on reasonable request.

## Code availability
Code used for the simulation of digital locks is available from the corresponding author upon reasonable request.

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

## Acknowledgements
The authors acknowledge support from RSF grant #19-12-00137. The authors are grateful to Denis Sukachev for careful reading and valuable discussion of the manuscript.

## Author contributions

V.S and N.K. conceived the experiment. K.Kh., V.S., and N.K. supervised funding and the workflow. D.T., A.G., D.P., D.M., and E.F. performed measurements. A.G. and D.T. contributed to the analysis and simulations. E.F., A.G., and D.T. produced figures. A.G. and N.K. drafted the manuscript. All authors discussed the results and contributed to the manuscript.

## Competing interests

The authors declare no competing interests.
