## [Peer Review File · Nature Communications]

Simultaneous bicolour interrogation in thulium optical clock providing very low systematic frequency shiftsREVIEWER COMMENTS

Reviewer #1 (Remarks to the Author):

Referee report on manuscript NCOMMS-21-10259-T

This paper reports experimental investigations related to an optical clock based on Thulium atoms confined in an optical lattice. It is shown that two hyperfine components of the clock transition can be interrogated simultaneously using a "bicolor" excitation and readout scheme. Since these components have exactly opposite quadratic Zeeman shifts, it is possible to accurately eliminate this systematic frequency shift of the clock. The ability to operate the clock with a rather large applied magnetic field is used to implement a measurement scheme that stabilizes the direction of the applied magnetic field in the presence of external perturbations. This stabilization in turn suppresses fluctuations of the tensorial Stark shift of the clock frequency.

The work reported in this paper is a significant step towards a practical Thulium optical lattice clock, which would be attractive in particular because the blackbody shift of the clock transition frequency at room temperature is much smaller than that of Sr and Yb lattice clocks. Most details of the presentation appear to be clear and adequate. However, I can recommend publication in Nature Communications only after a careful revision of the text according to the points of criticism noted in the following. Here the first point is most important, because it relates to one of the major claims of the paper.

Points of criticism:

(i) The discussion following Eq. (4) shows that for the operational values $\theta = \pi/2$ and $U = 100 E_r$, the tensor light shift of the clock transition is $\delta_{\text{nu}} \approx 70$ Hz, corresponding to a relative shift in the 10^{-15} range. The bias magnetic field direction stabilization shown in Fig. 4 is demonstrated under these experimental conditions, where θ is set to keep $|\delta_{\text{nu}}|$ at a local maximum. The actual size of $|\delta_{\text{nu}}|$ is proportional to the lattice light intensity so that long-term stabilization and reproducibility at the mHz level seems nearly impossible. It might be possible to set θ to the "magic angle" so that $\delta_{\text{nu}} = 0$, but this was not done, and it might be quite difficult to establish a corresponding stabilization protocol that is sufficiently accurate.

Thus, the claims in the Abstract ("The effect of the tensor lattice Stark shift can also be reduced to below 10^{-18} in fractional frequency units.") and in the Discussion (... we report extraordinarily low overall systematic frequency shifts ... /... we have experimentally demonstrated that the contribution of the tensor Stark shift ... can also be reduced to the 10^{-18} level by accurate online tuning ...) are unjustified and misleading. What was demonstrated instead was that the contribution of the magnetic field instability to the frequency instability of the thulium clock can be reduced to the 10^{-18} level.

(ii) On p.4, the term "asymmetric wave function" is used to explain why the differential tensor polarizability of the thulium clock transition is nonzero. This explanation is not understandable because it is not clear in which type of representation (or in which degree of freedom) the mentioned wavefunction asymmetry appears. It seems that the most obvious interpretation [loss of spatial (anti)symmetry] does not work. If the explanation based on symmetry properties can not be improved, I suggest to use the explanation that appears in a previous publication of the authors' group (Nat. Comm.

10, 1724 (2019)): "Unlike other systems used for optical clocks, the clock states in thulium have a large orbital momentum, which leads to tensor light shifts."

(iii) Throughout the text and in figure captions, the term "Allan variance" is used as a name for the quantity σ_y . For clarity, I would find it preferable that the authors adopt the standard usage of the terms "Allan variance" [for $(\sigma_y)^2$] and "Allan deviation" [for σ_y].

Reviewer #2 (Remarks to the Author):

The paper gives a description and implementation of an interrogation scheme used for a neutral thulium optical clock. The claim is made that this interrogation scheme makes the clock system almost free of systematic shifts at the 10⁻¹⁸ level. This is based on the claim that the only significant systematic is the second-order Zeeman shift and the tensor Stark shift, and that these are suppressed by the interrogation scheme. The clock system itself is an interesting one and some of the ideas are interesting but there are a number of issues with the paper that need to be addressed before it could be considered publishable. I discuss the problems in no particular order – mostly as they appear in the paper.

The first problem I have is one of insufficient referencing of relevant work. The interrogation technique proposed is essentially the hyperfine averaging technique [New Journal of Physics 17, no. 5 (2015): 053024] which has been demonstrated in different ways [Physical Review A, 94(5), 052512, Physical review letters, 124(8), 083202]. Here, the implementation uses simultaneous interrogation over two lines, but that does not change the fact that it is an averaging over hyperfine states. The authors have included ref[34] which mentions this approach but the reference is not given in the context of the interrogation technique proposed here. They should properly give a reference to the original work when introducing their proposed interrogation scheme. Additionally the comment on page 4 "This approach provides extremely low, compared to other widespread optical clocks, net systematic..." seems to be a deliberate wording to avoid acknowledging other systems that have equally good or better systematics that are not "widespread". Few, if any, optical clock systems can objectively be said to be widespread – all should be considered "under active investigation". Al⁺ (3 systems that I am aware of), In⁺ (1 system in Japan that has demonstrated clock operation and another under way in Germany) and Lu⁺ (1 system) all have low systematics – the latter in particular has lower systematics in all categories except maybe the probe Stark shift, for which auto-balancing or hyper-Ramsey techniques have been shown to be effective. What systems do the authors consider to be widespread on which their comparison is made?

The second problem I have is in over-stating a particular problem, namely assumptions about magnetic field noise behaviour between interrogations used in almost all other clocks. Page 5: "This fact results in full cancellation of the Zeeman shift without any assumptions about the magnetic field behavior between consecutive measurements." The truth is that this is a non-issue unless the magnetic field noise was limiting the clock stability, and, even then, it's mostly a problem of averaging time. To a large extent, it is no different than laser noise itself. Equally, one needs to be concerned about number fluctuations between interrogations, and this is particularly true for the joint interrogation sequence used here. Indeed, I argue that the authors would be better off not doing this joint interrogation but

rather interrogating sequentially over two lines as done in other systems (be it Zeeman pairs or hyperfine lines).

The third problem, and the reason the authors should not use this bicolor scheme, is in the corrections to Eq. S9 that come about from decays of one state to another. Those corrections are based on other measurements, which themselves are based on experimental parameters that can drift with time. If not correct or servoed in real time, they will result in an incorrect averaging via Eq 3 or S4 – effectively each transition will be incorrectly weighted. Even if servoed, it would seem to me an assumption is made on the lifetime or mechanisms by which the atoms flip states. Interrogating each line sequentially avoids this complication, and would not affect clock stability – as is the case for all other systems with magnetic field sensitivity at the levels given for this system. It is also not clear how collisions would affect this type of scheme. At the very least the authors should clarify how the accuracy of the measured lifetime and other factors that go into eq. S9 would factor into the clock performance (accuracy). An alternative interrogation technique that would also avoid the complications noted is the hyperfine averaging by dynamic decoupling [Physical review letters, 124(8), 083202], that uses microwave fields to cycle between the states. However, one may have to consider how this is to be done for both upper and lower states, but I think it should be a straight forward extension and easier to implement than the current approach. Note that hyperfine averaging does not cancel the tensor shift unless $I > J$, which is not the case here.

A fourth problem, is the rather confusing discussion on the tensor polarizability. It is not made clear that the definition of the magic wavelength for the lattice is inclusive of the tensor component such that only the angle needs calibrated. I had to discover that myself from reading ref [S3] when trying to determine what the tensor values actually were. As it is now, the reader can easily believe there is a 0.7Hz shift, which is 2.6×10^{-15} of the clock frequency, that needs to be somehow corrected for. The description in ref[S3] is much more enlightening. It also needs to be clarified how one disentangles the magic wavelength determination from the angle of the field, since the magic wavelength for this system is a function of that angle – indeed a misalignment of the angle of the field could just as easily be compensated by a change in the lattice wavelength. It would seem that the effectiveness of the servo is also tied to how well the magic wavelength matches the true magic wavelength at the angle of interest. Note that the claimed insensitivity of the magic wavelength (compared to Sr) does not appear to account for its sensitivity to magnetic field orientation for this system. The authors need to clarify how an error in the magnetic field angle made in the determination of the magic wavelength would manifest in the error budget pertaining to the interrogation angle specified here.

A fifth problem relates to the magnetic field servo. When servoing over a line by interrogating either side, the absolute step used is not important as the authors correctly point out. However, what is important, is that the step is the same in that the average of the two steps is zero i.e., they are equal and opposite. This is easily accomplished when stepping the frequency of a laser as that is set by a relatively low frequency rf oscillator, and hence accurately set relative to the clock frequency. Here the authors are jumping the magnetic field so how do they ensure the field shift is exactly equal and opposite for each step and not subject to overall drift itself or imperfect experimental conditions? Again, that would have to be calibrated and again subject to drifts and changes over time. Specifically, if the plus and minus change were different in magnitude by say 5%, there would be an assumption they were the same with a resulting incorrect change to the field alignment. Moreover, given the size of the shift, I would not be so quick to neglect higher order changes from drifts of the magnetic field along z.

Furthermore, as noted above, the achieved accuracy is seemingly dependent on how well one has gotten the magic wavelength correct (for the desired angle). There is no scientific data or methodology to support the notion that the field orientations are accurately set or a statement about what level needs to be achieved to obtain a given accuracy.

A sixth problem I have here is often a general one I see with optical lattice clocks. With the systematics purportedly at the low 10^{-18} , why is the clock integration only done for 20 minutes as per Fig 2? I cannot think of any good reason for optical lattice clocks to run for such a short time. There is no reason to not have them run for a day or even more. It would appear that the clock stability is around $1e^{-14}/\sqrt{\tau}$ so integrating down to the mid $1e^{-17}$ level should be reasonable and would help to support the claims being made.

A final minor point concerns the cancellation of the Zeeman shifts. It is trivial to show that the Zeeman shifts within each fine-structure level cancel under this scheme. In fact, following the omitted reference [New Journal of Physics 17, no. 5 (2015): 053024], it comes from the fact that the Zeeman Hamiltonian has zero trace for the $m=0$ states and hence the shifts must vanish no matter what the field is set to. However, this omits any Zeeman interaction with other levels. It is not clear to me if there is a significant Zeeman coupling to other levels as the atom is clearly not in the LS coupling regime. The authors should comment on this point as the accuracy of their measurement showing they cancel is insufficient for a high accuracy clock: I estimate $3.3e^{-17}$ uncertainty based on the quoted $-0.008(185)$ Hz/G² and operating field of 0.218G. One might argue this contradicts the title of the paper.

When claiming a clock is free of systematics at the 10^{-18} level, one should stop and consider what effects might be important. I have no idea what the quadrupole moment of this system is, but I would wonder what effect it might have. In an ion trap this is important not only in the DC but the AC as well [Phys. Rev. A 99, 022515, 2019] for which averaging techniques fail. It is likely not important for neutral atoms, but one has to remember that it is a field gradient that matters. Field curvatures have mm length-scale for ion-trap systems but for optical lattices they have a sub-micron scale. It is unclear from the information but likely the electric field strength is also much higher such that the scale could be many orders of magnitude higher. True, the frequency in ion traps is tens of MHz as opposed to hundreds of THz for a lattice. However, since the shift comes in quadratically on the interaction and hence field gradient, I would not simply assume it unimportant.

For the same reason I would ask what the M1 coupling is of the lattice beam to the clock transition, and, is there a significant ac Zeeman shift because of this? Based on the information given, I estimate the DC Zeeman shift due an M1 coupling between the upper and lower clock states to be 1.8mHz/G^2 . This is the dc quadratic Zeeman shift of the clock transition due to the Zeeman coupling between the (upper and lower) clock states. However, the ac shift from their optical lattice is substantially larger as it is relatively near resonance. With a lattice depth of $100E_r$ stated as typical in the paper, I estimate a shift of 0.77 Hz. This would be linear in the power of the laser so at the stated $300E_r$ it would be 2.3Hz. This is the shift due to the oscillating magnetic field of the laser. In principle this has a scalar and a tensor component and would contribute to an overall magic wavelength. Formulae are similar but the angle is different by 90° . It appears this effect has not been considered. Based on the geometry and the fundamental relation between the magnetic and electric field of the laser, it likely just offsets formulae slightly.

In addition to the above there are also some minor editorial notes I would make 1) Superlatives and hyperbole: I think we should put a stop to this in scientific writing. For example, the title “Extraordinarily low...”. Are they really so extraordinary? and by what objective measure? The tensor shift is noted to be 0.7Hz which is 2.6×10^{-15} of the clock frequency and hardly small given the difficulty in characterizing this beyond the 1% level. The validity or importance of the work would/should not change with the omission of the word extraordinary. The term is subjective and should be left for the reader to judge for themselves based on the facts presented. 2) Paragraph 1: it is stated that the authors proved the BBR shift was very low. This seems an odd expression. As an experimental result I would say it was demonstrated or confirmed. 3) Page 4, concerning the tensor shift it is stated that it is “significant at the requested level of uncertainty”. Again an odd term – who was it requested by? I think “desired” would be better or even “at the desired 10⁻¹⁸ level of uncertainty”.

We thank the reviewers for a careful reading and fair comments to the manuscript. The questions and comments are very helpful and beneficial. We have tried to answer all of the questions and have revised our manuscript correspondingly.

Following the Reviewers' comments we slightly shifted the focus of the manuscript (in the introduction and conclusions) to discussion of the synthetic frequency approach and comparison of simultaneous and sequential interrogation. In this paradigm, thulium optical clock serves as a test platform, where we experimentally demonstrate workability and advantages of the developed scheme. We have also changed the title of the manuscript to

“Simultaneous bicolour interrogation in thulium optical clock providing very low systematic frequency shifts”.

Please find below our response and the new version of the manuscript (the modifications are highlighted).

Referee #1

This paper reports experimental investigations related to an optical clock based on Thulium atoms confined in an optical lattice. It is shown that two hyperfine components of the clock transition can be interrogated simultaneously using a “bicolor” excitation and readout scheme. Since these components have exactly opposite quadratic Zeeman shifts, it is possible to accurately eliminate this systematic frequency shift of the clock. The ability to operate the clock with a rather large applied magnetic field is used to implement a measurement scheme that stabilizes the direction of the applied magnetic field in the presence of external perturbations. This stabilization in turn suppresses fluctuations of the tensorial Stark shift of the clock frequency.

The work reported in this paper is a significant step towards a practical Thulium optical lattice clock, which would be attractive in particular because the blackbody shift of the clock transition frequency at room temperature is much smaller than that of Sr and Yb lattice clocks. Most details of the presentation appear to be clear and adequate. However, I can recommend publication in Nature Communications only after a careful revision of the text according to the points of criticism noted in the following. Here the first point is most important, because it relates to one of the major claims of the paper.

Points of criticism:

1. - *The discussion following Eq. (4) shows that for the operational values $\theta = \pi/2$ and $U = 100E_r$, the tensor light shift of the clock transition is $\Delta\nu^t \approx 70$ Hz, corresponding to a relative shift in the 10^{-15} range. The bias magnetic field direction stabilization shown in Fig. 4 is demonstrated under these experimental conditions, where θ is set to keep $\Delta\nu^t$ at a local maximum. The actual size of $\Delta\nu^t$ is proportional to the lattice light intensity so that long-term stabilization and reproducibility at the mHz level seems nearly impossible. It might be possible to set θ to the “magic angle” so that $\Delta\nu^t = 0$, but this was not done, and it might be quite difficult to establish a corresponding stabilization protocol that is sufficiently accurate.*

Thus, the claims in the Abstract (“The effect of the tensor lattice Stark shift can also be reduced to below 10^{-18} in fractional frequency units.”) and in the Discussion (... we report extraordinarily low overall systematic frequency shifts ... we have experimentally demonstrated that the contribution of the tensor Stark shift ... can also be reduced to the 10^{-18} level by accurate online tuning...) are unjustified and misleading. What was demonstrated instead was that the contribution of the magnetic field instability to the frequency instability of the thulium clock can be reduced to the 10^{-18} level.

Response. We thank the Reviewer very much for bringing up this question. Of course, the frequency shift at 10 Hz level, which is linearly proportional to the laser power, is nearly impossible to determine and stabilize at the desired 1 mHz level. We acknowledge that we have written down Eq.(4) in incomplete form which led to the conclusion about huge size of the corresponding shift. We apologize for this obscurity and for omitting the scalar part of the polarizability in Eq. (4) which causes such misunderstanding.

Full optical lattice Stark shift of the clock transition must be written in the following form:

$$\Delta\nu = \Delta\nu^s + \Delta\nu^t = (\tilde{\alpha}^s + \tilde{\alpha}^t \times (3 \cos^2 \theta - 1)) \times (U/E_r). \quad (1)$$

For $\theta = \pi/2 + \delta\theta$, where $|\delta\theta| \ll 1$, this can be rewritten as:

$$\Delta\nu = (\tilde{\alpha}^s - \tilde{\alpha}^t) \times (U/E_r) + 3\tilde{\alpha}^t(U/E_r) \times \delta\theta^2 = \Delta\nu_{\pi/2} + \delta^t. \quad (2)$$

We use here the notation δ^t from Eq.(5) from the manuscript to refer to the θ -dependent part of the ac Stark shift. $\Delta\nu_{\pi/2}$ gives the clock transition frequency shift for $\theta = \pi/2$. At the wavelength near 1064 nm, to which we refer in the manuscript, this part becomes zero, thus fulfilling condition of the “magic” wavelength. Herewith, the non-zero tensor shift $\Delta\nu^t \approx +70$ Hz (mentioned by the Reviewer) is canceled by the scalar shift $\Delta\nu^s \approx -70$ Hz. When working

at the “magic” wavelength, the non-zero frequency shift may occur due to two reasons: (i) deviation of the lattice laser frequency from the exact “magic” value and (ii) due to the angular θ -dependency of the tensor polarizability, which is the part δ^t .

We apologize for the confusion caused by non-complete expression of the lattice light shift in the manuscript. We changed this part in the manuscript to be more specific and clear.

2. - On p.4, the term “asymmetric wave function” is used to explain why the differential tensor polarizability of the thulium clock transition is nonzero. This explanation is not understandable because it is not clear in which type of representation (or in which degree of freedom) the mentioned wavefunction asymmetry appears. It seems that the most obvious interpretation [loss of spatial (anti)symmetry] does not work. If the explanation based on symmetry properties can not be improved, I suggest to use the explanation that appears in a previous publication of the authors’ group (Nat. Comm. 10, 1724 (2019)): “Unlike other systems used for optical clocks, the clock states in thulium have a large orbital momentum, which leads to tensor light shifts.”

Response. We agree with the Reviewer that the explanation of a nonzero tensor polarizability through a large orbital momentum is much cleaner. We changed the manuscript accordingly:

However, the advantages of the $4f$ -shell clock transition come with a price: thulium atom has a relatively large magnetic dipole moment of $4\mu_B$ in the ground state, where μ_B is the Bohr magneton. ... Large orbital momentum results in a nonzero differential tensor polarizability, which is numerically small (0.2 a.u. at 1064 nm) ...

3. - Throughout the text and in figure captions, the term “Allan variance” is used as a name for the quantity σ_y . For clarity, I would find it preferable that the authors adopt the standard usage of the terms “Allan variance” [for σ_y^2] and “Allan deviation” [for σ_y].

Response. We thank the Reviewer for the suggestion and we have followed this recommendation.

Referee #2

The paper gives a description and implementation of an interrogation scheme used for a neutral thulium optical clock. The claim is made that this interrogation scheme makes the clock system almost free of systematic shifts at the 10-18 level. This is based on the claim that the only significant systematic is the second-order Zeeman shift and the tensor Stark shift, and that these are suppressed by the interrogation scheme. The clock system itself is an interesting one and some of the ideas are interesting but there are a number of issues with the paper that need to be addressed before it could be considered publishable. I discuss the problems in no particular order – mostly as they appear in the paper.

Response. We thank the Reviewer for fair and deep criticism as well as for the detailed comments. To our opinion, the key question addressed to the manuscript concerns the method of simultaneous interrogation of two clock transitions. The Reviewer asks about advantages which this method can bring compared to the sequential interrogation of two clock transitions. The synthetic frequency approach to cancel one of the systematic frequency shifts (e.g. the Zeeman shift or the ac Stark shift) is not novel and is already demonstrated in a number of experiments. Still, the simultaneous optical interrogation of two clock transitions have not been implemented yet (to our knowledge) and, as any new method, possesses certain advantages and disadvantages.

Following the Reviewer’s comment we slightly shifted the focus of the manuscript (in the introduction and the conclusions) to the discussion of the synthetic frequency and comparison of simultaneous and sequential interrogation. In this paradigm, thulium optical clock serves as a test platform, where we experimentally demonstrate workability and advantages of the developed scheme. Our new simulations show (see further for details) that there are two clear advantages of the simultaneous interrogation: (i) reduction of the Allan deviation by $\sqrt{2}$ and (ii) cancelling out the impact of magnetic field fluctuations on certain harmonics of the cycle frequency which can bring significant non-averaging shift of the clock transitions. The first issue (i) comes from the fact, that we reduce the duty cycle by the factor of 2 (today’s optical clocks are not limited by the projection noise). The second issue (ii) could be important for the non-laboratory environment, taking into account the fact that any magnetic field noise will contain spectral frequency components which are multiples of the interrogation/readout cycle rate. Remembering the difficulties which the Dick effect brings to the optical clocks (in some sense it possesses similar features), our method to cancel the impact of the low-frequency magnetic field fluctuations may be helpful for the future optical clocks generations drastically reducing the impact of the magnetic field noise. Today there are only a few atomic systems where simultaneous

interrogation can be implemented, but their number may increase in the future. Moreover, in contrast to the dynamic decoupling and other combined optical-rf methods, the suggested all-optical method allows to interrogate different transitions in optical domain which can be separated by hundreds of THz.

Further we answer the Reviewer's questions and place corresponding corrections in the text.

1. *The first problem I have is one of insufficient referencing of relevant work. The interrogation technique proposed is essentially the hyperfine averaging technique [New Journal of Physics 17, no. 5 (2015): 053024] which has been demonstrated in different ways [Physical Review A, 94(5), 052512, Physical review letters, 124(8), 083202]. Here, the implementation uses simultaneous interrogation over two lines, but that does not change the fact that it is an averaging over hyperfine states. The authors have included ref[34] which mentions this approach but the reference is not given in the context of the interrogation technique proposed here. They should properly give a reference to the original work when introducing their proposed interrogation scheme. Additionally the comment on page 4 “This approach provides extremely low, compared to other widespread optical clocks, net systematic...” seems to be a deliberate wording to avoid acknowledging other systems that have equally good or better systematics that are not “widespread”. Few, if any, optical clock systems can objectively be said to be widespread – all should be considered “under active investigation”. Al+ (3 systems that I am aware of), In+ (1 system in Japan that has demonstrated clock operation and another under way in Germany) and Lu+ (1 system) all have low systematics – the latter in particular has lower systematics in all categories except maybe the probe Stark shift, for which auto-balancing or hyper-Ramsey techniques have been shown to be effective. What systems do the authors consider to be widespread on which their comparison is made?*

Response. We thank the Reviewer very much for pointing out the insufficient referencing of other works and apologize for missing them in the text. Of course we are familiar with the dynamical decoupling techniques proposed for ion-based optical clocks (the work Phys. Rev. Lett., 124(8), 083202 was cited in our recent publication Fedorova et. al., Phys. Rev. A. 102, 063114 (2020)). It looks like the corresponding references were non-deliberately lost from the text during multiple revisions of the manuscript. We have included them into the Introduction section.

Regarding the second part of the Reviewer comment about the word “widespread” when stating “This approach provides extremely low, compared to other widespread optical clocks, net systematic...”. We meant the most commonly used systems based on Sr, Yb and Yb+. For Al⁺ clock the net systematic shift is currently almost 10⁻¹⁵ in relative units [Phys. Rev. Lett. 123, 033201 (2019)] with the leading contribution from the quadratic Zeeman shift. In⁺ and especially Lu⁺ ion optical clocks have much lower net systematic shifts, but they are still above 10⁻¹⁷ level. For In⁺ clock it is equal to -2.4×10^{-17} (with the BBR shift of -2.7×10^{-17}). For Lu⁺ the leading contribution comes from either the ac Stark shift from the probe field (46×10^{-17} for $^1S_0 \rightarrow ^3D_1$ transition) or the BBR shift (2.7×10^{-17} for the $^1S_0 \rightarrow ^3D_2$ transition). As the Reviewer noted, the ac Stark shift in Lu⁺ can be readily canceled using the auto-balancing or the hyper-Ramsey techniques, hereby the net shift for $^1S_0 \rightarrow ^3D_2$ transition is also expected to be at the low 10⁻¹⁸ level.

To avoid misunderstanding we removed this statement from the text.

2. *The second problem I have is in over-stating a particular problem, namely assumptions about magnetic field noise behavior between interrogations used in almost all other clocks. Page 5: “This fact results in full cancellation of the Zeeman shift without any assumptions about the magnetic field behavior between consecutive measurements.” The truth is that this a non-issue unless the magnetic field noise was limiting the clock stability, and, even then, it's mostly a problem of averaging time. To a large extent, it is no different than laser noise itself. Equally, one needs to be concerned about number fluctuations between interrogations, and this is particularly true for the joint interrogation sequence used here. Indeed, I argue that the authors would be better off not doing this joint interrogation but rather interrogating sequentially over two lines as done in other systems (be it Zeeman pairs or hyperfine lines).*

Response. We agree with the Reviewer that the magnetic field stability is usually not the main issue of concern in the most of the optical clocks. However, to our opinion, full cancellation of one of the important systematic effects without any assumptions about the magnetic field noise spectrum is a certain step forward. In the envisioned era of transportable clocks it would not be always possible to provide environmental conditions comparable to the laboratory ones. Thulium with its large sensitivity of the (single-) clock transition frequency to the magnetic field turns out to be the test system to justify the proposed approach of simultaneous interrogation. As we showed earlier [Golovizin et.al, Nat. Commun. 10 (2019) and Golovizin et. al., AIP Conference Proceedings, 2241, 020016 (2020)], in order to achieve 10⁻¹⁷ uncertainty of the second-order Zeeman shift one needs to provide magnetic field stability at the level of 10 μG which is a difficult task. Moreover, as we show below and in the additional materials, the magnetic field oscillations at a specific frequency (the multiple of the interrogation cycle rate) causes a systematic shift in the sequential interrogation scheme. Concerning fluctuations of the number of atoms between interrogation cycles, the pumping scheme used for preparation of the initial $m_F = 0$ states is insensitive to frequency or power fluctuations of the pumping radiation (discussed in Fedorova et. al., Phys. Rev. A. 102, 063114 (2020)). It is due to the large

frequency detuning from the resonances and saturation of the pumping process. Hereby, the fluctuations of the number of atoms are identical for both initial states and are less than 10% in our routine operation.

Additionally, as the Reviewer mentions, the noise in the magnetic field will manifest in degradation of the frequency stability. Although it is not a fundamental problem, in some important applications (e.g. the relativistic geodesy) lower instability would allow one to reach the target uncertainty in shorter averaging time.

We added discussion about the influence of fluctuating magnetic field in the Supplementary Material.

3. *The third problem, and the reason the authors should not use this bicolor scheme, is in the corrections to Eq. S9 that come about from decays of one state to another. Those corrections are based on other measurements, which themselves are based on experimental parameters that can drift with time. If not correct or servoed in real time, they will result in an incorrect averaging via Eq 3 or S4 – effectively each transition will be incorrectly weighted. Even if servoed, it would seem to me an assumption is made on the lifetime or mechanisms by which the atoms flip states. Interrogating each line sequentially avoids this complication, and would not affect clock stability – as is the case for all other systems with magnetic field sensitivity at the levels given for this system. It is also not clear how collisions would affect this type of scheme. At the very least the authors should clarify how the accuracy of the measured lifetime and other factors that go into eq. S9 would factor into the clock performance (accuracy). An alternative interrogation technique that would also avoid the complications noted is the hyperfine averaging by dynamic decoupling [Physical review letters, 124(8), 083202], that uses microwave fields to cycle between the states. However, one may have to consider how this is to be done for both upper and lower states, but I think it should be a straight forward extension and easier to implement than the current approach. Note that hyperfine averaging does not cancel the tensor shift unless $I > J$, which is not the case here.*

Response. The question raised by the Reviewer is very important. Indeed, mutual influence of the transitions seems to be the significant disadvantage of the proposed method and is the point of serious concern. The key idea that there is minimal influence of the 4-3 transition on the 3-2 transition frequency lays in locking the frequency of the 4-3 interrogation laser (as well as for 3-2) to the center of the corresponding resonance. Hereby there should be no asymmetric contribution if we interrogate the left and right slopes of the line sequentially. The influence of the 4-3 transition on the 3-2 comes only from the spontaneous decay from the $|c, F = 3, m_F = 0\rangle$ (the upper state of 4-3 transition) to $|g, F = 3, m_F \neq 0\rangle$ (other magnetic sublevels of the lower state of 3-2 transition) during the probe and readout periods. There are several effects which may change the coefficients in Eqs. S9-S11:

1. the upper clock level lifetime. It does not change during the experiment.
2. timing of the readout and repumping pulses. It is maintained constant with precision better than $1 \mu\text{s}$ while typical measurement intervals are 1 – 10 ms.
3. the repump $1.14 \mu\text{m}$ radiation frequencies. They are stabilized to the corresponding transitions with an error smaller than ~ 5 Hz. For a 4 ms-long repump pulse, the Fourier linewidth is ~ 200 Hz.
4. the repump $1.14 \mu\text{m}$ radiation power. It is monitored and kept constant with better than 5% constancy.

To provide a quantitative proof we have extended our simulations (also added to the Supplementary Material). We introduced the following fluctuations in the model:

1. Quantum projection noise using binomial distribution of the number of atoms ($\approx 10^4$ initially in $|g, F = 4, m_F = 0\rangle$ and $\approx 10^3$ in $|g, F = 3, m_F = 0\rangle$ states) at each step during each readout in the sequence.
2. Clock laser frequency noise: random-walk for frequency and white-noise for phase, similar or larger than experimentally observed.
3. Clock laser power noise: random-walk or white-noise, around 10% of total power, similar or larger than experimentally observed.
4. Fluctuations of the number of atoms in the lattice, dispersion equals 10% of total number.
5. Error in determination of the coefficients ξ_{c3} , ξ_{g3} and ξ_{c2} of population transfer, 10% dispersion.
6. Magnetic field fluctuations with different spectra.

Summarizing modeling results (see separate file with results), we did not observe any systematic frequency shift of the synthetic frequency in the simultaneous interrogation scheme. Regarding the frequency instability, the simultaneous interrogation scheme provides instability of 1.4 times smaller compared to the sequential one. The proportional and integral gains were adjusted for each locking scheme to achieve optimal performance. The factor of 1.4 ($\sqrt{2}$)

agrees with the one expected from the 2-times decrease of the measurement cycle time. Indeed, for the sequential measurements we spend only half of time to probe each of the transitions.

To study the impact of the magnetic field fluctuations we performed simulations for different power spectra of the magnetic field noise. For the Brownian noise and slow drifts we did not observe any significant difference compared to the other noise sources listed above (no systematic shifts, increase of the Allan deviation). However, if the magnetic field oscillates with the period equals to two full measurement intervals, we observe a systematic frequency shift for the sequential scheme. This effect readily follows from simple considerations (see illustration in the separate file with calculations), as in this case both transitions will be shifted in one direction which will not be compensated after building up the synthetic frequency. On the other hand, for the simultaneous method this effect completely cancels out. This is also true for magnetic noise frequencies that are odd-multiple of the full measurement cycle rate. The real spectrum of magnetic frequency noise may and probably does contain these harmonics. The simultaneous interrogation method offers straightforward solution of this envisioned problem.

We agree with the Reviewer that the existing dynamic decoupling techniques are also applicable in our setup and will produce similar synthetic (averaged) frequency. We motivate our decision to use all-optical method by two reasons. First, for Tm level scheme it is technically straightforward and requires only one additional ~ 200 MHz acousto-optic modulator, the rest is done by proper programming. The dynamic decoupling, in turn, requires installation of RF antennas and the use of high-power amplifiers. Second, one of the goals of the current work is to perform the proof-of-principle experiment using Tm clock as a test system. In the future, there can be other systems where frequency separation can reach optical range. In this case the dynamic decoupling would not be applicable, while the proposed scheme can be used e.g. to cancel out the ac Stark shift, quadratic Zeeman or BBR shift.

We add this discussion to the Supplementary Material of the manuscript.

4. *A fourth problem, is the rather confusing discussion on the tensor polarizability. It is not made clear that the definition of the magic wavelength for the lattice is inclusive of the tensor component such that only the angle needs calibrated. I had to discover that myself from reading ref [S3] when trying to determine what the tensor values actually were. As it is now, the reader can easily believe there is a 0.7 Hz shift, which is 2.6×10^{-15} of the clock frequency, that needs to be somehow corrected for. The description in ref[S3] is much more enlightening. It also needs to be clarified how one disentangles the magic wavelength determination from the angle of the field, since the magic wavelength for this system is a function of that angle – indeed a misalignment of the angle of the field could just as easily be compensated by a change in the lattice wavelength. It would seem that the effectiveness of the servo is also tied to how well the magic wavelength matches the true magic wavelength at the angle of interest. Note that the claimed insensitivity of the magic wavelength (compared to Sr) does not appear to account for its sensitivity to magnetic field orientation for this system. The authors need to clarify how an error in the magnetic field angle made in the determination of the magic wavelength would manifest in the error budget pertaining to the interrogation angle specified here.*

A fifth problem relates to the magnetic field servo. When servoing over a line by interrogating either side, the absolute step used is not important as the authors correctly point out. However, what is important, is that the step is the same in that the average of the two steps is zero i.e., they are equal and opposite. This is easily accomplished when stepping the frequency of a laser as that is set by a relatively low frequency rf oscillator, and hence accurately set relative to the clock frequency. Here the authors are jumping the magnetic field so how do they ensure the field shift is exactly equal and opposite for each step and not subject to overall drift itself or imperfect experimental conditions? Again, that would have to be calibrated and again subject to drifts and changes over time. Specifically, if the plus and minus change were different in magnitude by say 5%, there would be an assumption they were the same with a resulting incorrect change to the field alignment. Moreover, given the size of the shift, I would not be so quick to neglect higher order changes from drifts of the magnetic field along z. Furthermore, as noted above, the achieved accuracy is seemingly dependent on how well one has gotten the magic wavelength correct (for the desired angle). There is no scientific data or methodology to support the notion that the field orientations are accurately set or a statement about what level needs to be achieved to obtain a given accuracy.

Response. The forth and the fifth questions are associated with the magnetic field alignment, we address these points together.

We thank the Reviewer for pointing out the lack of clarity in our description of the magic wavelength, particularly when we speak about zeroing the sum of the scalar and the tensor differential polarizabilities. The first Reviewer addressed very similar question, and we hope we gave corresponding explanations and text changes in our response to the first comment of the Reviewer 1. We apologise for incomplete description which caused misunderstanding of both Reviewers.

Concerning other points mentioned by the Reviewer 2. After adding the second-order Zeeman shift to Eq. (2) (of this document) we obtain the following expression describing the shift of the clock transition frequency due to the

optical lattice and external magnetic field:

$$\Delta\nu = (\tilde{\alpha}^s - \tilde{\alpha}^t) \times (U/E_r) + 3\tilde{\alpha}^t(U/E_r) \times \delta\theta^2 + \beta B_0^2. \quad (3)$$

As the Reviewer noted, the magic wavelength depends on the angular misalignment $\delta\theta$. However, working at non-magic wavelength (when the first term is non-zero) does not influence the implemented algorithm of $\delta\theta$ determination and stabilization. It will give just a constant clock transition frequency shift at a fixed lattice trap depth. Hereby, we can stabilise magnetic field direction to $\theta = \pi/2$ independently on the detuning from the magic wavelength.

In the next paragraph we answer a question about the possible influence of non-equality of magnetic field steps in “+” and “-” directions. In the current setup, the magnetic fields for “+” and “-” steps are set by changing the current flowing through a single magnetic coil. As a current source we use one of the channels of LTC2662 current source (the five-channel, 300mA Current-Source-Output 12-Bit DACs from “Analog Devices”). According to its specifications, it possesses the integrated nonlinearity of less than 1 bit, or $2^{-12} \times 300 \text{ mA} = 73 \mu\text{A}$. We preliminary calibrated with a few different current-meters that its output deviates from a set value by less than 1 mA in the full range. Most likely this value is limited by the current-meters. In the “nominal” operation regime the drive current equals 150 mA, which is in the middle of the current source’s range. For “+” and “-” steps we add/subtract typically about $\sim 100 \text{ mA}$ (depends on the value of the bias field). Taking into account the results of calibration, we expect maximum difference of 2 mA for “+” and “-” steps, or 0.02 in relative units. This mismatch would result in θ stabilization offset of $\delta\theta_{\text{offs}} = d\Delta B_y / (2B_0) = d\Delta B_y / (2\Delta B_y) \times \Delta B_y / B_0 = 10^{-3}$, where we took into account that the magnetic field step $\Delta B_y \approx 0.1B_0$. This conservatively evaluated uncertainty provides the target angle error of no more than 10^{-3} . The mismatch caused by feasible coil deformation at different currents is also within this limit. In future, we plan to perform “+” and “-” steps by changing polarity of a separate coil, which would allow us to keep the current magnitude constant and use no assumptions about the linearity of the current source.

There is a problem mentioned by the Referee concerning coupling of the $\delta\theta$ -dependent part of the lattice shift and the second-order Zeeman shift. To address it more carefully, we consider situation, when both \vec{B}_x and \vec{B}_y are slightly misaligned (see illustration on Fig. 1) from their perfect orientation. We assume that \vec{B}_x has an angle of $\pi/2 + \alpha$ to the optical lattice polarization (ideally it is $\pi/2$), and \vec{B}_y has an angle of γ to the optical lattice polarization (ideally it is 0). Writing down the clock transition frequencies for “+” and “-” B_y steps (second and third terms in Eq. (3)), we find that equilibrium angle $\delta\theta_{eq}$ (i.e. the angle to which the stabilization process converges) equals

FIG. 1. Magnetic field direction stabilization.

$$\delta\theta_{eq} = \gamma \frac{-\eta}{1 - \eta}, \quad (4)$$

where $\eta = \frac{\beta B_0^2}{3\tilde{\alpha}^t(U/E_r)}$. In our experiments, $\eta \leq 0.1$ for typical $U = 300 E_r$ and $B_0 = 0.5 \text{ G}$. (maximum bias magnetic field among the measurements). Maximum misalignment of \vec{B}_y and the lattice polarization can be estimated to be no more than $\gamma = 0.1$ rad, which already can be easily detected by eyes. In this case, we have an error of no more than 0.01 rad, which is worse than we expected. In future experiments this issue can be addressed from two directions: (i) better alignment of \vec{B}_y , and (ii) use of not only 4-3 transition for \vec{B}_0 direction calibration, but both 4-3 and 3-2. Indeed, for 3-2 transition the only difference in Eq. (3) is “-” sign of β , thus

$$\delta\theta_{eq}^{comb} = \frac{\delta\theta_{eq}^{4-3} + \delta\theta_{eq}^{3-2}}{2} = \gamma \frac{-\eta^2}{1 - \eta^2}. \quad (5)$$

For the same $\eta = 0.1$ this already gives $\delta\theta_{eq}^{comb} = 0.01 \times \gamma$, that is less than targeted accuracy $\delta\theta = 10^{-3}$. Another possibility is to perform calibration measurements at larger optical trap depth. This would reduce η and lead to smaller $\delta\theta_{eq}$.

Misalignment of the magnetic field in z direction at the level of 0.1 rad does not effect the alignment procedure described above.

We included the above discussion to the Supplementary Material.

5. *A sixth problem I have here is often a general one I see with optical lattice clocks. With the systematics purportedly at the low 10-18, why is the clock integration only done for 20 minutes as per Fig 2? I cannot think of any good reason for optical lattice clocks to run for such a short time. There is no reason to not have them run for a*

day or even more. It would appear that the clock stability is around $1e-14/\sqrt{\tau}$ so integrating down to the mid $1e-17$ level should be reasonable and would help to support the claims being made.

Response. We agree with the Reviewer in this point. A day-long continuous operation of optical clocks is still challenging, with only a few groups who demonstrated it successfully. In our case there are two restrictions. Our laser systems, particularly cooling ones, can be continuously locked for only a few hours. Hereby, in most of the experiments we intentionally set the measurement duration of 0.5 hour to obtain similar datasets for different magnetic fields. We are now in the upgrade stage of our laser setup in order to improve its lock stability, and to allow us up to day-long continuous measurement. Besides that, we set up the next generation of thulium optical clocks for comparison.

We believe, that our relatively short-time experiments (of a few hours long) allowed us to demonstrate and study the simultaneous interrogation method which is the main focus of the manuscript. We agree with the Reviewer that accurate experimental proof of small systematic frequency shifts in Tm optical clock will require much longer and robust operation.

6. *A final minor point concerns the cancellation of the Zeeman shifts. It is trivial to show that the Zeeman shifts within each fine-structure level cancel under this scheme. In fact, following the omitted reference [New Journal of Physics 17, no. 5 (2015): 053024], it comes from the fact that the Zeeman Hamiltonian has zero trace for the $m=0$ states and hence the shifts must vanish no matter what the field is set to. However, this omits any Zeeman interaction with other levels. It is not clear to me if there is a significant Zeeman coupling to other levels as the atom is clearly not in the LS coupling regime. The authors should comment on this point as the accuracy of their measurement showing they cancel is insufficient for a high accuracy clock: I estimate $3.3e-17$ uncertainty based on the quoted $-0.008(185)$ Hz/G² and operating field of 0.218G. One might argue this contradicts the title of the paper.*

Response. The uncertainty of the measured quadratic coefficient for the synthetic frequency $\beta = -0.008(185)$ Hz/G² is currently limited by the measurement statistics. If one takes the coefficient at 1 standard deviation level $\beta_{1s,d} = -0.193$ Hz/G², this results in the synthetic frequency shift of -3.5×10^{-17} at $B_0 = 218$ mG as the Reviewer correctly pointed out. However, this value is only an upper limit of the quadratic Zeeman shift for the synthetic frequency from the measurements at our current statistics. It proves the consistency of the measurements and supports the theoretical conclusion, that for the synthetic frequency the effect must vanish. Indeed, according to our calculations in the Supplementary Material, as well as the consideration presented by the Reviewer, the hyperfine interaction should produce zero averaged quadratic Zeeman shift for two hyperfine $m_F = 0$ sublevels.

A small quadratic sensitivity mentioned by the Reviewer (also in the Note #8), comes from coupling of the clock levels to other electronic levels. We thank the Reviewer for pointing this issue out. It is rather difficult to calculate this coefficient accurately, but certain estimations can be done assuming the LS coupling. The latter is valid for the fine structure of the ground electronic Tm state (the lower and the upper clock levels), since the Lande g-factors calculated from LS model ($g_{J,g}^{LS} = 1.143$ and $g_{J,c}^{LS} = 0.857$) are close to the experimentally measured $g_{J,g} = 1.141189$ and $g_{J,c} = 0.8545$. The coupling of the clock levels themselves ($J = 7/2$ and $J = 5/2$) induces the quadratic Zeeman coefficient of $\beta_{1,14} = 3.8$ mHz/G². At $B_0 = 0.218$ G the synthetic frequency shift is estimated as $\Delta\nu_{\beta_{1,14}} = 0.7 \times 10^{-18}$. While this interaction increases the upper clock level energy, coupling to other atomic levels should decrease it. Thus we do not expect residual quadratic coefficient to be (much) larger than $\beta_{1,14} = 3.8$ mHz/G².

We added these estimations to the Supplementary Material.

7. *When claiming a clock is free of systematics at the 10-18 level, one should stop and consider what effects might be important. I have no idea what the quadrupole moment of this system is, but I would wonder what effect it might have. In an ion trap this is important not only in the DC but the AC as well [Phys. Rev. A 99, 022515, 2019] for which averaging techniques fail. It is likely not important for neutral atoms, but one has to remember that it is a field gradient that matters. Field curvatures have mm length-scale for ion-trap systems but for optical lattices they have a sub-micron scale. It is unclear from the information but likely the electric field strength is also much higher such that the scale could be many orders of magnitude higher. True, the frequency in ion traps is tens of MHz as opposed to hundreds of THz for a lattice. However, since the shift comes in quadratically on the interaction and hence field gradient, I would not simply assume it unimportant.*

Response. We agree with the Reviewer that there are a few sources of the clock transition frequency shift at the level of 10^{-18} besides discussed in this manuscript. In Golovizin et.al, Nat. Commun. 10 (2019) we have analysed them, including those mentioned by the Reviewer. We showed that the quadrupole shift is negligible since the gradient of the electric field in the optical lattice oscillates at optical frequencies thus averages out. Similar situation is with the magnetic field of the blackbody radiation, where the estimated shift is less than 0.1 mHz, or 10^{-18} at room temperature.

The Reviewer also pointed out to the collisional shift, which is present in optical lattice clocks. We are strongly interested in theoretical estimations of this shift, however this is a rather complicated task. We look forward to

collaborate with theoreticians (e.g. Mariana Safronova) to investigate this problem, as well as other aspects like higher-order polarizabilities. We hope that the progress in thulium optical clock demonstrated in the present manuscript would stimulate theoretical estimations of these issues, which require separate study. We do not expect critical problems from the collisions due to the following reasons. (i) the differential polarizability of the clock transition in thulium is very small, thus the electric-type interaction should not lead to a significant frequency shift. (ii) atoms are in the $m_F = 0$ state. (iii) With high-power lasers at 1064 nm (the magic wavelength) we are able to produce large-volume traps to minimise atomic density. Of course, there are also certain concerns about quadrupole an isotropic instructions which were studied in Tm by the group of W. Ketterle and J. Doyle.

8. *For the same reason I would ask what the M1 coupling is of the lattice beam to the clock transition, and, is there a significant ac Zeeman shift because of this? Based on the information given, I estimate the DC Zeeman shift due an M1 coupling between the upper and lower clock states to be 1.8mHz/G². This is the dc quadratic Zeeman shift of the clock transition due to the Zeeman coupling between the (upper and lower) clock states. However, the ac shift from their optical lattice is substantially larger as it is relatively near resonance. With a lattice depth of 100Er stated as typical in the paper, I estimate a shift of 0.77 Hz. This would be linear in the power of the laser so at the stated 300Er it would be 2.3Hz. This is the shift due to the oscillating magnetic field of the laser. In principle this has a scalar and a tensor component and would contribute to an overall magic wavelength. Formulae are similar but the angle is different by 90°. It appears this effect has not been considered. Based on the geometry and the fundamental relation between the magnetic and electric field of the laser, it likely just offsets formulae slightly.*

Response. We are very grateful to the Reviewer for bringing up the question. The differential polarizability of the clock levels from the lattice magnetic field equals $\Delta\alpha_{1,14} = 0.19 \times 10^{-3}$ a.u at 1064 nm. This value (with addition of contribution from other M1 and E2 transitions) contributes to the lattice Stark shift in a different way than E1 polarizability: the induced frequency shift is proportional to the magnetic field amplitude $B^2 \sim \sin^2(kr)$ (or electric field gradient $|dE/dr|^2 \sim \sin^2(kr)$ for E2) rather than to the electric field $E^2 \sim \cos^2(kr)$ of the optical lattice. This results in shifts which are non-linear with light intensity [a good example is Ushijima et.al., Phys. Rev. Lett. 121, 263202 (2018)]. Our calculated value of $\Delta\alpha_{1,14}$ leads to the similar coefficient $\tilde{\alpha}^{qm}$ as in the case of the Sr optical lattice clocks. For the latter, a sub 10^{-17} shift of the clock transition frequency from the optical lattice was experimentally demonstrated. According to our estimations, contribution of the other M1 and E2 transitions should be much smaller than $\Delta\alpha_{1,14}$. Meanwhile, the sensitivity to the lattice laser frequency and hyperpolarizability in our case are much smaller than for Sr lattice clocks. The fluctuations of the tensor polarizability shift can be kept below 10^{-18} , as we discussed in our response to Note #4. This should provide the lattice shift of less than 10^{-17} . Again, rigorous theoretical calculations and analysis of higher-order polarizability terms would provide more accurate estimations.

We have changed corresponding parts of the manuscript, where we discuss the optical lattice shifts, and added estimations of the higher-order lattice shift to the Supplementary Material.

9. *I addition to the above there are also some minor editorial notes I would make*

1) Superlatives and hyperbole: I think we should put a stop to this in scientific writing. For example, the title "Extraordinarily low...". Are they really so extraordinary? and by what objective measure? The tensor shift is noted to be 0.7Hz which is 2.6×10^{-15} of the clock frequency and hardly small given the difficulty in characterizing this beyond the 1% level. The validity or importance of the work would/should not change with the omission of the word extraordinary. The term is subjective and should be left for the reader to judge for themselves based on the facts presented.

Response.

We are grateful to the Reviewer for pointing out some improper hyperbolization, and following the Reviewer recommendation to be more specific we changed the manuscript title to

"Simultaneous bicolor interrogation in thulium optical clock providing very low systematic frequency shifts"

Summing up all the discussion above, we believe that we have clarified a number of issues concerning the systematic shifts in Tm raised by the Reviewers and demonstrated certain advantages of the simultaneous all-optical interrogation scheme:

1. We experimentally showed that the quadratic Zeeman shift is less than 3×10^{-17} for the synthetic frequency, this value is an upper limit set by statistical uncertainty of the experiment. Theoretical calculations predict that this shift is less than 10^{-18} for magnetic field of 0.2 G.
2. Clock transition frequency shift from the non-zero tensor polarizability cancels by the scalar part of the polarizability. The lattice Stark shift is small (sub- 10^{-17}) and arises from the higher order terms of polarizability

and hyperpolarizability, similarly to other lattice optical clocks. At a moment it seems to be the dominating systematic effect in our system. Fluctuations of the angle-dependent part of the tensor lattice Stark shift can be kept below 10^{-18} . The magic wavelength at around 1064 nm is much less sensitive to the lattice laser frequency changes compared to the situation in other optical lattice clocks.

3. The blackbody radiation shift is $2.3(1.1) \times 10^{-18}$ at room temperature.
4. the ac Stark shift from the probe field is negligible.
5. the quadrupole shift averages to zero owing to high oscillating frequency of the optical field.

We agree that there are many more shifts that need to be taken into account (e.g. collisional shift), however they are typically at 10^{-18} level for other neutral systems. We do not see the reasons why they could be larger in thulium.

As the Reviewer correctly mentioned, there are a few elements which have low net systematic shift, e.g. In+ and especially Lu+. One can say that this fact even positively emphasizes the advantages of Tm inner-shell transition in thulium, keeping in mind that this is a neutral system. Among the other neutral-atoms-based optical clock, the lowest total systematic shift is at the level of 10^{-16} at room temperature (Hg) or at cryogenic temperatures (for Sr).

Doing this research, we mostly focused on envisioned transportable systems which are today considered as the key element for e.g. relativistic geodesy. Ion-based systems (Yb+, Ca+) require long averaging time, which is critical for this purpose. Transportable neutral Sr clock systems are in turn very delicate (laser wavelengths, secondary cooling, 813 nm lattice frequency noise, etc) and require accurate consideration of systematic shifts.

Beginning our studies of Tm we did not fully believe that Tm could be one of the most robust neutral systems in respect to different kind of external perturbations. Our recent research shows, that we can straightforwardly cancel out most of the systematic shifts to low 10^{-18} level which, together with an extremely convenient magic wavelength at 1064 nm, makes the system probably one of the easiest neutral-atoms based system to implement for transportation. In our present Tm system the least robust part seems to be the first-stage cooling (410 nm), which can be upgraded straightforwardly.

2) Paragraph 1: it is stated that the authors proved the BBR shift was very low. This seems an odd expression. As an experimental result I would say it was demonstrated or confirmed.

Response. We agree with the Reviewer here and have changed the manuscript accordingly.

This is especially important for transportable and envisioned space-based optical clocks. We previously confirmed that the $1.14 \mu\text{m}$ inner-shell magnetic dipole transition in neutral thulium possesses a very low blackbody radiation shift compared to other neutrals.

3) Page 4, concerning the tensor shift it is stated that it is "significant at the requested level of uncertainty". Again an odd term – who was it requested by? I think "desired" would be better or even "at the desired 10-18 level of uncertainty".

Response. We have changed the manuscript accordingly.

Large orbital momentum results in a nonzero differential tensor polarizability, which is numerically small (0.2 a.u. at 1064 nm) but significant at the desired \$10^{-18}\$ level of uncertainty.

REVIEWERS' COMMENTS

Reviewer #1 (Remarks to the Author):

My points of criticism have been adequately taken into account in the revised manuscript. Therefore I recommend publication.

Reviewer #2 (Remarks to the Author):

Review of the bicolor interrogation of a Tm clock

The authors have made significant changes to the paper to address most of the concerns raised. Overall I think the Tm clock system is an interesting one and something for the clock community to keep an eye on. There are a few points, that I will make for the record. To what extent the authors choose to address these is up to them.

1. I don't think it is overly clear on the relation of the magic wavelength and the field alignment. In their rebuttal they state that not working at the magic wavelength will "give just a constant clock transition frequency shift at a fixed lattice trap depth". That is exactly the point, and that should be clarified in the paper. To be explicit, I outline how I imagine setting up a Tm clock.

To find the magic wavelength, one would measure the clock difference at high and low lattice depth (power) and make them as equal by adjusting the wavelength. Ideally the low power would be the clock operating power and the high power as large as possible. If the high power was 10 times larger than the low power, a clock comparison at 10^{-17} would reduce the uncertainty from the lattice Stark shift to 10^{-18} at the operating power. But the tensor polarizability makes the magic wavelength dependent on the angle between the B-field and the lattice polarization i.e. there is no unique magic wavelength. So the questions are

- (a) If I run the proposed magnetic field servo on each clock at high and low power, will I uniquely identify the $\theta = \pi/2$ magic wavelength?
- (b) If there is any bias in the magnetic field servo from step imbalances or field alignment, will I at least find a magic wavelength albeit for a different θ ?

I believe the answer to both questions is yes. For the latter I would hope/or assume further running of the servo would maintain reasonable alignment.

Note that the manner in which the magic wavelength is found makes the slope of the polarizability curve redundant. There is no 1000-fold advantage over Sr: the increased slope for Sr means the magic wavelength can be determined that much more accurately, nonlinear effects notwithstanding. The resulting contribution to the overall clock uncertainty at the operating power of the lattice is then limited by the accuracy of the clock comparison and the maximum power one can operate at when making the magic wavelength determination, not the slope of the curve.

2. In my previous report I had overlooked the spatial dependence of the M1 and E2 couplings, but there

is a question that arises from Table I of the supplemental. I find it remarkable that $\tilde{\alpha}^{qm}$ in table I can be similar in magnitude to Sr, given that the Tm clock transition is M1 and the lattice is detuned by a mere 18 THz.

To be specific, the shift of the Tm clock transition due to an oscillating magnetic field is given by

$$\hbar\Delta\omega_c = \frac{\frac{1}{3} + \frac{23}{294}(3\cos^2\theta_m - 1)\mu_B^2\langle B^2\rangle}{1 - (\omega/\omega_c)^2} \hbar\omega_c$$

where ω_c is the (angular) clock frequency, ω the frequency of the oscillating magnetic field, θ_m is the angle between the oscillating field direction and the applied magnetic field, and we have used the approximation that the reduced matrix element for the M1 clock transition is $\sqrt{24/7}\mu_B$. This expression is the M1 analogue of the E1 expression. Taking α^{qm} defined by

$$h\Delta f_c = -\frac{1}{2}\alpha^{qm}\langle B^2\rangle = -\frac{\alpha^{qm}}{\alpha^{E1}} \frac{E_r}{c^2} \frac{U}{E_r}$$

would give

$$\tilde{\alpha}^{qm} = \frac{\alpha^{qm}}{\alpha^{E1}} \frac{E_r}{c^2\hbar},$$

and hence $\tilde{\alpha}^{qm} \approx 15$ mHz. The definition of α^{qm} differs from [Phys. Rev. Lett. **121**, 263202 (2018)] in scale but is consistent with the authors version of α^{E1} for the trap depth (S5), such that the ratio should be correct. Relative to the clock frequency this is 25 times larger than the quoted value for Sr - and about what I would expect. Note that the expression given above for Tm has the correct dc limit and the matrix element gives the correct lifetime. Some indication of where these numbers actually come from would be useful for the more interested reader.

3. In reference to hyperfine averaging, I don't think it is appropriate to say the bicolor excitation scheme is similar to the dynamic decoupling demonstrated in [44]. The interrogation scheme in [44] is a realization of the hyperfine averaging technique proposed in [40]. Similarly, the bicolor excitation scheme presented in this work is a realization of the hyperfine averaging technique proposed in [40]. Making a distinction between interrogating sequentially or simultaneously in the implementation of hyperfine averaging is splitting hairs.
4. Whether you interrogate sequentially or simultaneously also makes no difference to the stability. You do not gain a factor of $\sqrt{2}$ as indicated on page 9. With simultaneous interrogation you have half the

number of atoms that you would have when operate sequentially so it makes no difference. If you have an imbalance in the populations you will actually lose stability being limited by that with the least atoms. This ignores magnetic field noise considerations, which, in all practical circumstances, is likely not important.

I. MAGNETIC FIELD SENSITIVITIES

These points are no longer an issue given the current way the work is presented, but given the comments made in the rebuttal it is worthwhile to clarify a few things.

I think it is misleading to compare the overall magnetic field shift for a clock when speaking of advantages between clocks. Firstly, the magnetic field can be very accurately calibrated and tracked during clock operation so the contribution of the magnetic field uncertainty to the error budget is negligible relative to the contribution arising from the quadratic Zeeman coefficient, α_z . I believe this to be true for all clocks. Secondly, I am aware of only one practical way to measure α_z and that is through clock comparison. In a nutshell, a comparison between two clocks at B_0 and B_f gives a quadratic Zeeman coefficient of

$$\alpha_z = \frac{\delta f_z}{B_f^2 - B_0^2}, \quad (1)$$

where δf_z is the frequency difference of the two clocks (corrected for all other systematics). The uncertainty in α_z is then determined by the systematic uncertainty in the two clocks, σ , as the magnetic field uncertainty does not contribute. Hence, at the operating field, B_0 , the contribution to the fractional frequency uncertainty budget due to the measurement precision of α_z is

$$\frac{\delta f}{f_c} = \frac{B_0^2}{B_f^2 - B_0^2} \frac{\sigma}{f_c}. \quad (2)$$

Hence one is limited only by the clock performance, which would almost certainly be statistical (projection noise limited), and the maximum field one can reliably operate at. Importantly, one is not limited by the overall size of the shift. This is intuitively reasonable: if the coefficient is very large, the shift is very large and α_z maybe accurately measured - this is the case for Al^+ as quoted by the authors. If the coefficient is very small, it cannot be accurately determined - this is the case for Tm . Admittedly, if the argument could be made that the quadratic Zeeman shift for Tm is dominated by the M1 clock transition itself, then one could measure the lifetime accurately to determine the matrix element and hence have an independent assessment of the quadratic shift coefficient - this would be a nice result in fact.

When it comes to magnetic field noise, the authors use an oscillating magnetic field to demonstrate the advantage of their technique. This is exactly the Dick effect applied to the magnetic field. I would argue it is much less severe than the Dick effect applied to the laser noise. Maybe this is much more of a problem for Tm given the short lifetime, but this is hardly an advantage over other clocks. In Al^+ , the relevant sensitivity to magnetic field noise is in the g-factors and that leads to something like 4 kHz/G. For Lu^+ (3D_1) the relevant number comes from the quadratic shift of the most sensitive state: at an operating field of 1 G it is ~ 40 Hz/G. These are only problematic for interrogation times well beyond what is possible in Tm . Magnetic field oscillations that are coherently periodic with the interrogation would be indicative of technical problems with equipment: eddy currents from a switching field would be something I would look for.

Dear Editor, thank you very much for organizing the next round of reviews of our manuscript and for forwarding the reviewers' comments which show their positive evaluation of our work. We are glad to hear that both reviewers found our response to their first-round comments satisfying. Please find below our comments to fresh notes from the second reviewer.

Referee #2

The authors have made significant changes to the paper to address most of the concerns raised. Overall I think the Tm clock system is an interesting one and something for the clock community to keep an eye on. There are a few points, that I will make for the record. To what extent the authors choose to address these is up to them.

1. *I don't think it is overly clear on the relation of the magic wavelength and the field alignment. In their rebuttal they state that not working at the magic wavelength will "give just a constant clock transition frequency shift at a fixed lattice trap depth". That is exactly the point, and that should be clarified in the paper. To be explicit, I outline how I imagine setting up a Tm clock. To find the magic wavelength, one would measure the clock difference at high and low lattice depth (power) and make them as equal by adjusting the wavelength. Ideally the low power would be the clock operating power and the high power as large as possible. If the high power was 10 times larger than the low power, a clock comparison at 10^{-17} would reduce the uncertainty from the lattice Stark shift to 10^{-18} at the operating power. But the tensor polarizability makes the magic wavelength dependent on the angle between the B-field and the lattice polarization i.e. there is no unique magic wavelength. So the questions are*

- (a) *If I run the proposed magnetic field servo on each clock at high and low power, will I uniquely identify the $\theta = \pi/2$ magic wavelength?*
- (b) *If there is any bias in the magnetic field servo from step imbalances or field alignment, will I at least find a magic wavelength albeit for a different θ ?*

I believe the answer to both questions is yes. For the latter I would hope/or assume further running of the servo would maintain reasonable alignment. Note that the manner in which the magic wavelength is found makes the slope of the polarizability curve redundant. There is no 1000-fold advantage over Sr: the increased slope for Sr means the magic wavelength can be determined that much more accurately, nonlinear effects notwithstanding. The resulting contribution to the overall clock uncertainty at the operating power of the lattice is then limited by the accuracy of the clock comparison and the maximum power one can operate at when making the magic wavelength determination, not the slope of the curve.

Response. Yes. Since the angle between the magnetic field and the lattice polarization does not depend on the lattice depth, the stabilization procedure of θ to $\pi/2$ may be performed only when the lattice depth is large since the error offset (eq. 14 in the Supplementary Material) becomes smaller. As we also showed in the Supplementary, one can readily achieve error in θ stabilization to $\pi/2$ of less than 1 mrad, that corresponds to 10^{-18} clock transition shift and thus should resolve any issues with the θ -dependency of the magic wavelength. We introduced to the Supplementary Material one paragraph with this discussion.

The reviewer absolutely correctly says that residual uncertainty from the magic wavelength determination basically depends only on the uncertainty of clock comparison rather than on the polarizability slope. However, having smaller slope of the differential polarizability is advantageous due to the reduced demands on the control of frequency of the lattice laser (reversely proportional to the slope magnitude). In the Discussion section in manuscript we refer only to this sensitivity: "... it provides small sensitivity of the clock transition frequency to the lattice wavelength ..."

2. *In my previous report I had overlooked the spatial dependence of the M1 and E2 couplings, but there is a question that arises from Table I of the supplemental. I find it remarkable that $\bar{\alpha}^{qm}$ in table I can be similar in magnitude to Sr, given that the Tm clock transition is M1 and the lattice is detuned by a mere 18 THz.*

To be specific, the shift of the Tm clock transition due to an oscillating magnetic field is given by

$$\hbar\Delta\omega_c = \frac{\frac{1}{3} + \frac{23}{294}(3\cos^2\theta_m - 1)}{1 - (\omega/\omega_c)^2} \mu_B^2 \langle B^2 \rangle \hbar\omega_c \quad (1)$$

where ω_c is the (angular) clock frequency, ω the frequency of the oscillating magnetic field, θ_m is the angle between the oscillating field direction and the applied magnetic field, and we have used the approximation that the reduced

matrix element for the M1 clock transition is $\sqrt{24/7} \mu_B$. This expression is the M1 analogue of the E1 expression. Taking α^{qm} defined by

$$h\Delta f_c = -\frac{1}{2}\alpha^{qm} \langle B^2 \rangle = -\frac{\alpha^{qm} E_r}{\alpha^{E1}} \frac{U}{c^2 E_r} \quad (2)$$

would give

$$\tilde{\alpha}^{qm} = \frac{\alpha^{qm} E_r}{\alpha^{E1} c^2 h} \quad (3)$$

and hence $\tilde{\alpha}^{qm} \approx 15$ mHz. The definition of $\tilde{\alpha}^{qm}$ differs from [Phys. Rev. Lett. **121**, 263202 (2018)] in scale but is consistent with the authors version of α^{E1} for the trap depth (S5), such that the ratio should be correct. Relative to the clock frequency this is 25 times larger than the quoted value for Sr - and about what I would expect. Note that the expression given above for Tm has the correct dc limit and the matrix element gives the correct lifetime. Some indication of where these numbers actually come from would be useful for the more interested reader.

Response.

We thank the reviewer for providing calculations of the M1 polarizability at 1064 nm caused by the clock transition. Our current theoretical knowledge does not allow to deduce Eq(1), so if the reviewer could share some link to the source, we would be extremely grateful. Hereby, we can only comment on how we did our estimations. We used eqs (4-6) from [Phys. Rev. A **94**, 022512 (2016)] and substituted probability of the clock transition (M1) instead of E1 transitions. Please let us know if this approach is incorrect. Doing this, we found $\alpha^{M1} = 1.9 \times 10^{-4}$ a.u. This value is indeed 40 times larger than the differential M1 polarizability for Sr atom [Phys. Rev. Lett. **120**, 063204]. However, in Sr the dominant contribution to α^{qm} comes from the E2 transitions. Additionally, since the recoil energy in Tm is $E_r \approx h \times 1$ kHz (which is smaller than in Sr) we end up with a similar value of $\tilde{\alpha}^{qm}$ to the one in Sr.

3. *In reference to hyperfine averaging, I don't think it is appropriate to say the bicolor excitation scheme is similar to the dynamic decoupling demonstrated in [44]. The interrogation scheme in [44] is a realization of the hyperfine averaging technique proposed in [40]. Similarly, the bicolor excitation scheme presented in this work is a realization of the hyperfine averaging technique proposed in [40]. Making a distinction between interrogating sequentially or simultaneously in the implementation of hyperfine averaging is splitting hairs.*

Response. In the part of the text to which the reviewer is referencing, we just wanted to note that our approach allows one to average over the clock transitions between different hyperfine components similarly to the hyperfine averaging scheme from [New J. Phys. **17** 053024 (2015)].

To resolve any misunderstanding we replaced that part with the following:

We note here, that recently implemented dynamic decoupling scheme [40,44] allows one to average out the second-order Zeeman shift and quadrupole electric shift.

4. *Whether you interrogate sequentially or simultaneously also makes no difference to the stability. You do not gain a factor of $\sqrt{2}$ as indicated on page 9. With simultaneous interrogation you have half the number of atoms that you would have when operate sequentially so it makes no difference. If you have an imbalance in the populations you will actually lose stability being limited by that with the least atoms. This ignores magnetic field noise considerations, which, in all practical circumstances, is likely not important.*

Response. This is absolutely true when one is limited by the QPN. However in our and most neutral-atoms-based optical clocks, instability is yet limited by other factors, typically by the laser noise.

The imbalance of the population of two initial states is one of our concerns. For now, we do not see any signs of stability downgrade for the 3-2 clock transition (where we have 10 times lower atoms number) compared to 4-3 transition, since the clock laser noise dominates. In future, one can equalize the number of atoms in $|g, F = 4, m_F = 0\rangle$ and $|g, F = 3, m_F = 0\rangle$ by applying a short laser pulse coupling $|J = 7/2, F = 4\rangle \rightarrow |J = 9/2, F = 4\rangle$ transition (another transition of our second stage cooling) to initially pump half of all atoms to $|g, F = 3\rangle$. To implement this, we need only one AOM or EOM with ~ 1.5 GHz drive frequency.

5. MAGNETIC FIELD SENSITIVITIES. *These points are no longer an issue given the current way the work is presented, but given the comments made in the rebuttal it is worthwhile to clarify a few things.*

I think it is misleading to compare the overall magnetic field shift for a clock when speaking of advantages between clocks. Firstly, the magnetic field can be very accurately calibrated and tracked during clock operation so the contribution of the magnetic field uncertainty to the error budget is negligible relative to the contribution arising from the quadratic Zeeman coefficient, α_z . I believe this to be true for all clocks. Secondly, I am aware of only one practical way to measure α_z and that is through clock comparison. In a nutshell, a comparison between two clocks at B_0 and B_f gives a quadratic Zeeman coefficient of

$$\alpha_z = \frac{\delta f_z}{B_f^2 - B_0^2} \quad (4)$$

where δf_z is the frequency difference of the two clocks (corrected for all other systematics). The uncertainty in α_z is then determined by the systematic uncertainty in the two clocks, σ , as the magnetic field uncertainty does not contribute. Hence, at the operating field, B_0 , the contribution to the fractional frequency uncertainty budget due to the measurement precision of α_z is

$$\frac{\delta f_c}{f_c} = \frac{B_0^2}{B_f^2 - B_0^2} \frac{\sigma}{f_c} \quad (5)$$

Hence one is limited only by the clock performance, which would almost certainly be statistical (projection noise limited), and the maximum field one can reliably operate at. Importantly, one is not limited by the overall size of the shift. This is intuitively reasonable: if the coefficient is very large, the shift is very large and α_z maybe accurately measured - this is the case for Al^+ as quoted by the authors. If the coefficient is very small, it cannot be accurately determined - this is the case for Tm. Admittedly, if the argument could be made that the quadratic Zeeman shift for Tm is dominated by the M1 clock transition itself, then one could measure the lifetime accurately to determine the matrix element and hence have an independent assessment of the quadratic shift coefficient - this would be a nice result in fact.

When it comes to magnetic field noise, the authors use an oscillating magnetic field to demonstrate the advantage of their technique. This is exactly the Dick effect applied to the magnetic field. I would argue it is much less severe than the Dick effect applied to the laser noise. Maybe this is much more of a problem for Tm given the short lifetime, but this is hardly an advantage over other clocks. In Al^+ , the relevant sensitivity to magnetic field noise is in the g -factors and that leads to something like 4 kHz/G. For Lu^+ (3D_1) the relevant number comes from the quadratic shift of the most sensitive state: at an operating field of 1 G it is ~ 40 Hz/G. These are only problematic for interrogation times well beyond what is possible in Tm. Magnetic field oscillations that are coherently periodic with the interrogation would be indicative of technical problems with equipment: eddy currents from a switching field would be something I would look for.

Response. Indeed, in order to account for i.e. quadratic Zeeman shift one needs to know (1) the coefficient α_z , as the reviewer discusses above, but also (2) amplitude of the magnetic field, since the error can be written as

$$\delta f_c = \delta\alpha_z B_0^2 + 2\alpha_z B_0 \delta B_0. \quad (6)$$

Hereby, the smaller α_z one has, the softer requirements on the magnetic field control. Ideally, as we expect in Tm, the shift itself would be below 10^{-18} for bias magnetic field of ~ 0.1 G, so there would be no need for its calibration. In this case we would only need to constrain shift magnitude by the way the reviewer described. The suggestion about comparing the residual quadratic Zeeman coefficient with the one calculated from the clock levels coupling is very interesting and we plan to finalize this research in the near future. We thank the reviewer for this suggestion.

The reviewer's arguments in the last paragraph are absolutely correct. But when moving to field applications with transportable systems, insensitivity to different shifts, including those caused by technical noise, is beneficial. In the manuscript, we highlight advantages of the proposed interrogation scheme in Tm being insensitive to fluctuations of the magnetic field specifically when we discuss its application for transportable clocks.